

# Impacts of climate mitigation strategies in the energy sector on global land use and carbon balance

Kerstin Engström[1], Mats Lindeskog[1], Stefan Olin[1], John Hassler[2], and Benjamin Smith[1]

[1] Department of Physical Geography and Ecosystem Science, Lund University, Lund, 22362, Sweden.
[2] Institute for International Economic Studies, Stockholm University, Sweden.

*Correspondence to*: Kerstin Engström (kerstin.engstrom@nateko.lu.se)

**Abstract.** Reducing greenhouse gas emissions to limit climate change-induced damage to the global economy and secure the livelihoods of future generations requires ambitious mitigation strategies. The introduction of a global carbon tax on fossil fuels is tested here as a mitigation strategy to reduce atmospheric $CO_2$ concentrations and radiative forcing. Taxation of fossil fuels potentially leads to changed composition of energy sources, including a larger relative contribution from bioenergy. Further, the introduction of a mitigation strategy reduces climate change-induced damage to the global economy, and thus can indirectly affect consumption patterns and investments in agricultural technologies and yield enhancement. Here we assess the implications of changes in bioenergy demand as well as the indirectly caused changes in consumption and crop yields for global and national cropland area and terrestrial biosphere carbon balance. We apply a novel integrated assessment modelling framework, combining a climate-economy model, a socio-economic land-use model and an ecosystem model. We develop reference and mitigation scenarios based on the Shared Socio-economic Pathways (SSPs) framework. Taking emissions from the land-use sector into account, we find that the introduction of a global carbon tax on the fossil fuel sector is an effective mitigation strategy only for scenarios with low population development and strong sustainability criteria (SSP1 "Taking the green road"). For scenarios with high population growth, low technological development and bioenergy production the high demand for cropland causes the terrestrial biosphere to switch from being a carbon sink to a source by the end of the 21st century.

## 1 Introduction

Combating climate change is one of the greatest challenges of the 21st century. Currently the world is on an emission pathway that approaches the highest of the four Representative Concentration Pathways (RCPs; Fuss et al., 2014; Peters et al., 2012). If emissions are not drastically curbed within the next few decades a global average surface warming of 3.7° C to 4.8° C compared to pre-industrial levels and more frequent extreme weather events will be the likely consequence (IPCC, 2014). Such profound changes in the climate system are strongly linked with changes in the terrestrial biosphere. During the last 250 years a share of carbon dioxide ($CO_2$) emissions has been taken up by the terrestrial biosphere, thus referred to as a



carbon sink (Canadell and Schulze, 2014). The future of the terrestrial carbon sink is highly uncertain (Ahlström et al., 2012; Ciais et al., 2013) and depends on processes and feedbacks involving the carbon cycle, nutrient dynamics, disturbances such as wildfires, and land use, the latter driven by the demand for land to grow biomass for food, feed and fuel. Land-use and land cover change (LULCC) are themselves drivers of climate change. During 1750-2012 deforestation and agricultural

management are estimated to have contributed 30% to anthropogenic $CO_2$ emissions, while this share decreased to 10% in the period 2000-2012, mainly due to decreasing deforestation rates (Canadell and Schulze, 2014; Ciais et al., 2013). Including other greenhouse gases (GHG, e.g. methane and nitrous oxide), LULCC and agriculture were responsible for 21% of total emissions in 2010 (Tubiello et al., 2015), the remainder stemming from the combustion of fossil fuels and industrial processes.

Mitigation strategies are designed to slow down or limit climate change with the purpose of decreasing negative impacts on society. One manifestation of mitigation measures would be higher gross world product (GWP), which would allow higher consumption and higher investments in e.g. agricultural production. The transition towards carbon neutral energy sources and reduction in overall energy use are key elements of proposed mitigation strategies. One proposed mitigation measure is to introduce a global carbon tax which creates incentives to reduce overall energy use and to replace fossil fuels with

renewable energies, including bioenergy. Bioenergy can be derived from energy crops or residues from other land uses such as forestry (Haberl et al., 2010). An inevitable effect of increased bioenergy use will be an increasing demand for land (Wise et al., 2009; Hassler and Sinn, under review), the displacement of lands formerly used for traditional agriculture, or the extension of land use into areas occupied by natural ecosystems. Moreover, not all bioenergy systems lead to net emissions reductions, especially if the carbon debt (carbon released when land was cleared initially for bioenergy production) is

included (Fargione et al., 2008).

The demand for food and feed is dependent on societal and technological development, e.g. population growth, changes in diets and yield management. Socio-economic scenarios describe the joint evolution of different aspects of development. Here we apply a novel Integrated Assessment Modelling (IAM) framework to explore the impacts of a global carbon tax on fossil fuels on global land use and land-atmosphere carbon exchange, within the wider context of different socio-economic

scenarios of the Shared Socio-economic Pathways (SSPs; O'Neill et al., in press) framework. We quantify the indirect and direct effects of ambitious energy mitigation strategies for each SSP on food consumption, yield development and cropland. Further we study the impact of mitigation-derived changes in climate and cropland on the terrestrial carbon balance to finally address the question whether or not a global sectorial carbon tax on the energy system is an effective strategy to mitigate climate change at the global scale.



## 2 Methods

### 2.1 Reference and mitigation scenarios in the IAM framework

We developed two sets of scenarios based on the socio-economic developments described in the SSP scenario framework (Fig. 1). The SSPs outline five plausible pathways that global societal development could follow in the 21st century and are

characterized by the development of key elements, such as population, equity, economy, trade, lifestyle, policies, technology and energy intensity (O'Neill et al., in press). The SSPs do not take into account potential impacts of climate change or new climate policies and can thus be considered reference scenarios with respect to climate change (O'Neill et al., 2013). The first set of scenarios used in this study is strictly based on the SSPs and thus consists of five reference scenarios where no new climate polices were considered. A second set of scenarios complemented each reference scenario with a mitigation

strategy consistent with relevant aspects with the reference scenario storyline. The mitigation measure considered was the introduction of a global carbon tax on fossil fuels targeting overall reductions in energy use and the replacement of fossil fuels with renewable energies, including bioenergy. Carbon taxes are generally regarded as an effective economic incentive to reduce greenhouse gas emissions and lead to less volatility in emissions prices than quantity restrictions as in carbon trading schemes (Golosov et al., 2014; Hassler et al., in press). Instead, the tax can be set equal to the expected damage of a

marginal unit of emissions allowing market participants to take these damages into account when making economic decisions.

Mitigation through carbon sequestration, e.g. by afforestation schemes or carbon capture and storage technology, is not considered. However, the five mitigation scenarios encompass strategies that are assumed to affect the speed and strength of technological growth of energy production technologies and infrastructure, alongside the level of global carbon tax imposed

in the mitigation scenarios. Instead of defining a target (e.g. global average temperature increase of less than 2°C) and designing a climate policy that is likely to achieve this target (as in the Shared Policy Assumptions, SPAs, see Kriegler et al., 2014), we chose to assign mitigation strategies that are consistent with the challenges for mitigation implied by each SSP. This approach tests the maximum capacity of mitigation for each SSP in the energy sector and results will indicate if more efforts are needed to reach certain targets. The SSPs have varying challenges for mitigation due to their different key

characteristics (Fig. 1), as for example the high energy demand in SSP5 "Taking the highway" and the slow technological change in SSP3 "A rocky road" (O'Neill et al., in press). The differences in non-climate policies and institutions contribute as well to varying challenges for mitigation, as e.g. the environmental awareness and effective institutions in SSP1 "Taking the green road" decrease the challenge for mitigation compared to e.g. SSP5 "Taking the highway" (O'Neill et al., in press). The SSPs are used to parameterize processes such as the development of energy technologies and available labour in a

climate-economy model (Fig. 2).

The climate-economy model calculates the social cost of carbon emissions from the energy sector, equivalent with the optimal carbon tax and the damage to GWP. Damages are determined as a function of simulated mean global temperature in turn driven by the endogenously determined emission path. Thus, the climate-economy model is used to create emission



scenarios, assess damage to GWP and simulate renewable and fossil energy demands (for details see Sect. 2.2.1). Further, the SSPs provide input data for population and economic development and characterize technological change and consumption patterns, required as input to a socio-economic land-use model. The land-use model reconciles demand for food, feed and bioenergy implied by the scenario assumptions with the biophysically-determined supply (productivity) of

these commodities per unit land area on a country-by-country basis, and translates this into cropland changes (for details see Sect. 2.2.2). The land-use model uses yield scenarios, which are the result of calculating the distances of the emission scenarios (indicated by Δ in Fig. 1) to the RCPs and using these as inverse weights to create yield time series (Appendix A5; Engström et al., 2016a) based on simulated cropland productivity from an ecosystem model (for details see Sect. 2.2.3).

The land-use model uses the scenario-specific damage to GWP and yield data to explore the indirect impact of damages of

GWP on food consumption and yield development, as well as the direct impact of bioenergy demand on cropland area in each country. Resulting cropland changes are downscaled to grid cell level (see Appendix A7) and the impact of cropland changes, as well as the mitigation-derived reduced climate change, on the terrestrial carbon balance is estimated by the ecosystem model.

### 2.2 Component models

### 2.2.1 Climate-economy model

The climate-economy model is a dynamic general-equilibrium model that predicts the joint evolution of the global climate and economy, operating at the global scale (Golosov et al., 2014). In the model, forward-looking agents decide how much to consume and save. Firms make production decisions, taking prices and taxes as given. The use of three different types of energy, namely oil, coal and clean energy (renewables and nuclear, free of fossil carbon emissions), is determined as a

market outcome such that supply equals demand at all points in time. The fact that markets are modelled explicitly makes the model different from the most popular economic models used to study climate change and therefore well suited to study how different policies, e.g., carbon taxes at different levels, affect the market outcome.

Golosov et al. (2014) show that the common assumption of a quadratic damage function in combination with the logarithmic relation between atmospheric $CO_2$-concentration (proportional to the atmospheric carbon pool) and forcing implies that the

marginal damage flow elasticity is approximately constant. Specifically, a marginal unit of airborne carbon has an approximately constant percentage impact on GWP independent of the $CO_2$ concentration. Golosov et al. (2014) calculate the damage elasticity ($\gamma$) to $2.38 \times 10^{-5}$ per airborne GtC implying that an extra GtC in the atmosphere reduces the flow of GWP by $2.38 \times 10^{-3}$ percent. Finally, Golosov et al. (2014) show that the optimal carbon tax is proportional to GWP with a proportionality factor given by the product of the expected value of $\gamma$ and the carbon duration $D$ defined as in Eq. (1):

$$D \equiv \int_0^\infty e^{-\rho t} \psi(t) dt \qquad (1)$$

where $\rho$ is the rate at which future welfare is discounted and $\psi(t)$ is the share of a unit of carbon emissions that remain airborne $t$ units of time after it was emitted.





The model endogenously solves for the use of the three types of energy and carbon emissions. Key parameters determining the emissions paths are the rate of growth in the efficiency of producing coal ($A_{2,g}$) and clean energy ($A_{3,g}$, where clean energy includes nuclear energy and renewables) and the elasticity of substitution ($se$) between these types of energy in the production of final goods. A higher growth rate in the efficiency of coal (clean energy) production leads, as long as other

variables are held constant, to slower price growth and faster growing use of coal (clean energy). The sensitivity of this mechanism is determined by how substitutable the different types of energy are in the production of final goods. Baseline assumptions about these parameters are listed in Table1 and are assumed to be amenable for scenario specific developments. In the different scenarios we will make different assumptions about the parameters in Table 1. The technological growth rates are allowed to differ across the SSP's but to a less extent also between the reference and mitigation scenarios. Strictly

speaking, the economic model does not contain a mechanism whereby policy makers could affect these growth rates. However, it would be straightforward to allow the growth rates to be determined by how R&D efforts are allocated between different uses. This would make it possible for policy makers to partly control relative and absolute technology growth rates without important changes in the model's predictions (see e.g., Hassler et al., under review, for an example of endogenous energy related technical change). A similar argument can be made regarding the substitution elasticity where it is assumed

that policy makers can facilitate a transition to a cleaner energy production by slightly increasing the elasticity. However, in all cases, the elasticity is fairly close to unity.

### 2.2.2 Land-use model and coupling to the climate-economy model

The land-use model PLUM (Parsimonious Land-Use Model) simulates changes in cropland coverage on the basis of changes in cereal, meat and milk consumption and changes in cereal yield in 168 countries (Engström et al., 2016b). Calculations of

food demand are dependent on population and economic development and are described by statistical relationships revealed by historical country-level statistics from reported data (FAOSTAT, 2016). The coefficients characterizing these relationships are used as scenario parameters. Scenario parameters are based on the SSP characteristics as previously described in Engström et al. (2016a). Population, economic development and the share of urban population on total population are input to PLUM and are used as provided by the SSP database (SSP-Database, 2015). Changes in expected

production are simulated via a global rule-based trade mechanism. The expected cereal production together with cereal yield is used to simulate changes in cereal land. During the simulation period it is assumed that actual national yields in PLUM are changing towards potential yield, simulated for multiple RCP × GCM climate trajectories by the ecosystem model LPJ-GUESS (see Sect. 2.2.3 and Engström et al., 2016a), depending further on each scenario's technological growth, economic development and technology transfer. Finally, changes in total cropland are assumed to be proportional to changes in cereal

land, using the actual proportions of cereal land to total cropland in 2000 (Engström et al., 2016b). In previous applications of PLUM (Engström et al., 2016a) the static feed ratio (assumption as to how much of the consumed meat is produced from cereal feeds vs. grazing) was identified as a cause for underestimation of cropland demand for scenarios with meat-rich diets.



Here we assumed the feed ratio to increase proportional to increases in consumption of animal products if the initial feed ratio is very low, restricted by a scenario specific maximum for the feed ratio (*feedRatioCap*; see Appendix A1).

The simulated damage to GWP from the climate-economy model was downscaled to country level gross domestic product (GDP), adjusting the shares covered by high, medium and low income countries, depending on the level of social equity of

each SSP (*equity*, see Appendix A2). This formulation reflects the assumption that low income and vulnerable countries would not receive much support by high income countries to deal with the consequences of climate change in the case of low equity. The downscaling approach reinforces the pattern of decreasing economic inequalities across low, medium and high income countries for high equity scenarios, while it slows down the decreasing income gap for scenarios with low equity (see Table A. 1, Appendix A2).

The output of clean energy from the climate-economy model was used to derive bioenergy scenarios, which were then translated into explicit cropland demands for bioenergy in PLUM. To arrive at the bioenergy scenarios we assumed that the shares of different clean energy sources (nuclear and renewables, i.e. hydro, wind, solar and bioenergy) projected by the World Energy Outlook (WEO) scenarios (current policy, new policy and 450ppm; Appendix A3; OECD/IEA, 2012) are representative for scenarios with high, medium and low challenges towards mitigation in the SSP challenge space (Fig. 1).

The resulting projections of bioenergy are assumed to be produced from a range of available sources, such as industrial waste, forestry residues, agricultural by-products and energy crops. Energy crops in the WEO scenarios are defined as "those (crops) grown specifically for energy purposes, including sugar and starch feedstocks for ethanol (corn, sugarcane and sugar beet), vegetable-oil feedstocks for biodiesel (rapeseed, soybean and oil palm fruit) and lignocellulosic material (switchgrass, poplar and miscanthus)" (OECD/IEA, 2012). In PLUM we only explicitly model the share of bioenergy

produced from energy crops (excluding lignocellulosic feedstocks), which was 3% in 2000 (OECD/IEA, 2012). The future contribution of energy crops to total bioenergy potential is highly uncertain depending on assumptions as to available croplands and yield development, but considering sustainability constraints has been suggested to range from 30-50% in 2050 (Haberl et al., 2010). Lignocellulosic feedstocks are expected to play a major role in future bioenergy production, but as they are excluded here we assume a lower contribution of energy crops to total bioenergy of maximum 15% in 2100

(*shareBEcr*, see Appendix A4). The modelled bioenergy production occurs here predominantly on abandoned cropland, but if this is not available it is expanded into remaining natural vegetation (grasslands and or forest; see Appendix A4). Bioenergy production is assumed to be predominantly produced in countries with large bioenergy production today as well as countries with sufficient remaining natural vegetation (in cases where bioenergy cannot be produced on abandoned cropland). Furthermore, bioenergy production efficiency is assumed to increase at different rates depending in the scenario,

bound by the upper range of values reported today (*efficiencyBE* see Appendix 4; Börjesson and Tufvesson, 2011).

### 2.2.3 Ecosystem model, downscaling cropland and the terrestrial carbon balance

The managed land version of the dynamic vegetation model LPJ-GUESS (Smith et al. 2001; Lindeskog et al., 2013), was used to simulate cereal yields (wheat, maize, millet and rice) as input to PLUM as in Engström et al. (2016a). The



simulations capture the impact of climate change on yield developments on a $0.5 \times 0.5°$ global grid through changes in precipitation, temperature patterns and $CO_2$ concentration (derived from the RCPs, see Engström et al., 2016a). The initial difference between actual and potential yield was established by scaling the simulated yield to actual and potential yield from Mueller et al. (2012) for the year 2000. The scaling factor was used throughout 2000-2100. For use in PLUM, actual

and potential yields were aggregated from grid cells to country level using area fractions from the MIRCA2000 data set (Portmann et al., 2010). The SSP-RCP matrices for the reference and mitigation scenarios (Appendix A5) were used to weight simulation outputs with the four RCPs.

The country-level changes in cropland area simulated with PLUM were applied to a base map of current land cover (cropland and grassland) extent on a $0.5 \times 0.5°$ global grid (Hurtt et al. 2011). A downscaling algorithm was used to

disaggregate land cover from country to grid cell level based on a weighted combination of proximity to existing cropland and suitability based on simulated potential crop productivity, capturing both expansion and contraction of current land cover extent. A detailed description of the downscaling algorithm is provided in Appendix A7.

To estimate the combined effects of biophysical drivers and land use change on biospheric terrestrial carbon balance, we applied LPJ-GUESS globally on the $0.5 \times 0.5°$ grid of the downscaled land use data, simulating natural vegetation (also

encompassing forest), cropland and pasture and the dynamic transitions between these land cover types (Lindeskog et al., 2013). Natural vegetation in the model emerges as the result of growth and competition for light and soil resources among woody plant individuals and a herbaceous understorey in each of a number (5 in this study) of replicate patches (0.1 ha), representing stochastic variation in the history of vegetation evolution (succession) following disturbance across the landscape of a simulated grid cell (Smith et al., 2014). Multiple plant functional types (PFTs) co-occur and compete within

each patch, and age/size classes are distinguished for trees, capturing effects of stand demography on biomass accumulation and turnover. C-N interactions were taken into account, following Smith et al. (2014). Pasture is represented by herbaceous ($C_3$ or $C_4$ grass) PFTs, harvested yearly. Cropland is represented by wheat and maize following the implementation of Olin et al. (2015), with relative areas aggregated from the MIRCA2000 data for the year 2000 (Portmann et al., 2010), and taken to represent all $C_3$ (wheat) and $C_4$ (maize) crops globally, including energy crops. Irrigation was applied for cropland,

according to historical global irrigation data for the year 2000 (Portmann et al., 2010), nitrogen fertilization according to historical data for the period 1901-2006 (Zaehle et al., 2010). Tillage and cover-crops were cropland management options considered in all simulations (Olin et al., 2015).

For the historical period (1700-2000), we used cropland, pasture and natural area fraction data for 1700-2000 from Hurtt et al. (2011), global atmospheric $CO_2$ concentrations for 1850-2000 from the CMIP5 archives (Taylor et al., 2012) and nitrogen

deposition data for 1850-2000 from Lamarque et al. (2011). For future simulations (2001-2100) climate input to the ecosystem model simulations was provided by bias-corrected fields of mean monthly temperature, precipitation and incoming shortwave radiation for the atmosphere-ocean general circulation model (GCM) IPSL-CM5A-MR (IPSL, Dufresne et al., 2013). IPSL was selected as it simulates changes that are located in the middle of the ensemble spanned by all GCMs (Ahlström et al., 2012), which was confirmed by running three additional GCMs (GFDL-CM3 (Donner et al., 2011),



MIROC5 (Watanabe et al., 2010) and MRI-CGCM3 (Yukimoto et al., 2012)) for medium LUC (SPS2m) and RCP4.5. The climate model forcing fields were bias corrected relative to observed historical climate from the CRU TS3.0 dataset (Mitchell and Jones, 2005) and downscaled to the grid of the land use data, following Ahlström et al (2012). Carbon pools in the model were initialized to equilibrium with the early-20th century historical climate by a 500 year "spin-up" forced by

prescribed 1700 land cover, 1850 atmospheric $CO_2$ concentration and nitrogen deposition, 1901 nitrogen fertilizer applications for cropland, and detrended monthly climate time series for 1850-1879, cycled repeatedly.

Carbon cycle simulations were performed for the scenario period 2001-2100, separately for the reference and mitigation scenarios for each SSP. Time-varying cropland-area fractions simulated by PLUM were applied as anomalies relative to baseline (2000) land use from the Hurtt et al. (2011) product, downscaled from country to grid cell level, as described above.

Separate simulations were performed for each RCP × GCM combination; nitrogen deposition data were taken from Lamarque et al. (2011) for the relevant RCP. Relative crop type distribution, irrigation, nitrogen application (after 2006) and tillage intensity were kept constant at modern (2006) levels. Model outputs were aggregated to grid cell averages for each SSP, weighting simulations according to the probabilistic mapping of each SSP to each RCP shown in the Appendix (Table A.3).

### 2.4 Parameterizing the models for the reference and mitigation scenarios

#### 2.4.1 Parameter settings for the climate economy model

The parameterization of the climate economy model was oriented at a selection of key elements specifying the SSPs (O'Neill et al., in press), listed in Table 2. Additionally to these key elements we included the second axis of the challenge space, the challenge for adaptation, to derive a sensitive parameterization of the damage elasticity factor ($\gamma$) for each scenario. In the

climate economy model, the damage factor $\gamma$ describes the impact of emissions and climate change on GWP. Higher $\gamma$ means that in mitigation scenarios emissions will need to be decreased substantially to avoid anticipated higher damages.

The challenges for adaptation are low for SSP1 "Taking the green road" and SSP5 "Taking the highway", medium for SSP2 "Middle of the road" and high for SSP3 "A rocky road" and SSP4 "A road divided", which was translated into quantitative values for the damage factor $\gamma$, see Table 3. Golosov et al. (2014) show that $\gamma$ of $5 \times 10^{-5}$ per airborne GtC fairly well

approximates a middle-range climate sensitivity of 3°C (i.e. a doubling of the atmospheric carbon pool leads to a 3°C increase in global mean temperature) and a damage function following Nordhaus (2007). Acknowledging the limited evidence for the calibration of $\gamma$, we choose this to represent a relatively benign situation and also use higher gammas. For the scenarios we therefore chose 5, 10 and $15 \times 10^{-5}$ per airborne GtC to represent low, medium and high damage factors.

The parameter settings for $\gamma$ are assumed to be equal in the reference and mitigation scenarios per SSP. For the reference

scenarios no carbon tax is assumed ($\tau=0$), see Table 3.

As for the reference scenarios, the SSPs form the basis of the mitigation scenarios. In addition to introducing carbon taxes, mitigation strategies could, as described above, encompass the following changes relative to the reference scenario: (1)



reduced growth of extraction efficiency of coal; (2) increased growth of efficiency of green technologies; and (3) The increased substitution elasticity in order to further stimulate the production of green (=clean) energy.

All these changes should be consistent with the storylines of the SSPs and with the challenges for mitigation (high, medium, low) of the respective SSP. Parameter choices are shown in Table 3. The level of the carbon tax $\tau$ (Table 1) for the mitigation

scenarios is a fraction of the optimal carbon tax ($\tau$). We assumed that the mitigation strategies for scenarios with low challenges to mitigation (SSP1 "Taking the green road" and SSP4 "A road divided") imply an optimal carbon tax ($\tau$ =1). The optimal carbon tax reduces emissions to the level where the costs of avoiding emissions and the cost of avoided damages are at equilibrium. The mitigation strategy for SSP2 "Middle of the road" (medium challenge to mitigation) consists of 30% of the optimal carbon tax ($\tau$ =0.3). For scenarios with high mitigation challenges (SSP5 "Taking the highway" and SSP3 "A

rocky road") we assumed that the mitigation strategy is 10% of optimal carbon tax ($\tau$ =0.1). This reflects the belief that political problems associated with introducing a global tax may lead to a tax substantially lower than the optimal (see Appendix A6).

### 2.4.2 Parameter settings for land use model

For the parameterization of the land use model PLUM for the five reference scenarios we relied on the parameterization as in

Engström et al., (2016a), except for the scenario parameters newly introduced in this study, as e.g. the maximum feed ratio *feedRatioCap* (Table 4, Appendix A1). The second new scenario parameter is *equity* which directly relates to the human development key element "Equity" of the SSPs as described in O'Neill et al. (in press). Equity is described to be high for SSP1 "Taking the green road" and SSP5 "Taking the highway"; medium for SSP2 "Middle of the road" and SSP4 "A road divided"; and low for SSP3 "A rocky road" (O'Neill et al., in press). In PLUM, *equity* steers which downscaling approach

for damage to GWP is used, see Table 4.

The implementation of bioenergy in PLUM introduced two additional scenario parameters, *shareBEcr*, which describes the increase of bioenergy produced from energy crops, and *efficiencyBE*, which accounts for efficiency improvements in energy conversion. The share of bioenergy that was produced from energy crops in the period 2000-2010 was 3% and was assumed to increase to up to 6% in 2100 for the reference scenarios with sustainability focus (SSP1 "Taking the green road") and

reference scenarios which, at least partly, are strongly reliant on local energy sources (SSP3 "A rocky road" and SSP4 "A road divided"). For the fossil fuel focused SSP5 "Taking the highway" no changes from the initial values for the bioenergy scenario parameters were made, neither for reference or mitigation scenario. For the mitigation version of SSP1 "Taking the green road" the share of bioenergy crops was assumed not to increase further than in the reference scenario, due to the fact that the use of cropland for bioenergy production and its effect on sustainability can be negative in some cases.

By contrast, for SSP4 "A road divided", which, as SSP1 "Taking the green road", has a low challenge to mitigation but less focus on sustainability, it was assumed that bioenergy production from energy crops would increase to up to 9% in 2100. SSP3 with its high challenge to mitigation was assumed to keep the share of bioenergy crops as in the reference scenario, but increase the efficiency in bioenergy conversion slightly.



## 3 Results

### 3.1 Global energy scenarios, atmospheric carbon, damage to GWP and cropland development

With no mitigation, global energy use increases steeply for all scenarios, least for SSP1 "Taking the green road", and spans a range of 1000-2000 EJ by 2100 (Fig. 3). The predominant energy sources across the reference scenarios differ. While in

SSP5 "Taking the highway", fossil fuel dominates, in all other reference scenarios, renewable energies and bioenergy contribute to the rising energy demand, especially for the sustainability-oriented SSP1 "Taking the green road". The introduction of a global carbon tax on fossil fuels as mitigation strategy effectively reduces the energy consumption to around 1000 EJ in 2100 for all scenarios (Fig. 3). However, the contributions of fossil fuels, renewable energies and bioenergy to total energy supply differ greatly across the mitigation scenarios and reflect the varying global carbon taxes.

Assumed political support for carbon taxes in scenarios with low challenges for mitigation (see Sect. 2.4.1) leads to high carbon taxes (SSP1 "Taking the green road": 115 US$ per ton carbon at 2010 GWP and SSP4 "A road divided": 344 US$ per ton carbon at 2010 GWP), decreasing the contribution of fossil fuel to total energy use to around 10% in 2100. By contrast, for SSP5 "Taking the highway" the global carbon tax is only 11 US$ per ton carbon and fossil fuels remain the main energy source even in the mitigation scenario.

The concentration pathway of atmospheric carbon for SSP5 "Taking the highway" marks the upper end of the simulated concentration pathways, though it remains lower than the very steep trajectory of the RCP8.5 radiative forcing scenario (Fig. 4, panel a). Only SSP1 "Taking the green road" achieves an atmospheric carbon pathway close to RCP4.5 for the reference case, while the remaining scenarios are all clustered around RCP6.0 (Fig. 4, panel a). The introduction of a global carbon tax and the subsequent reduction of energy use and transition towards renewable energies yield considerably lower

concentration pathways for the mitigation scenarios. SSP1 "Taking the green road" and SSP4 "A road divided" approach RCP2.6, while SSP2 "Middle of the road" and SSP3 "A rocky road" are between RCP2.6 and RCP4.5, leaving SSP5 "Taking the highway" with the highest mitigation concentration pathway (close to RCP6.0, Fig. 4, panel a).

If not mitigated, climate change causes damage to GWP by up to 12 % in 2100 (SSP3 "A rocky road" and SPP4 "A road divided", Fig. 4, panel b). For scenarios with low and medium challenges for adaptation the damage is 3-9% of GWP in

2100. Climate change mitigation strategies reduce the impact to below 8% damage to GWP in 2100 for all scenarios (Fig. 4, panel b). The largest reduction of damage occurs for SSP4 "A road divided" (from 12% to 5% of GWP, Fig. 4 and Fig. 5) where the low challenges for mitigation enable a strong reduction in emissions, while the high challenges for adaptation make such reduction very desirable. Similar reasoning explains why the global carbon tax in SSP4 "A road divided" is significantly higher than in the other scenarios. The impacts of the avoided damage in the mitigation scenarios on food

consumption, yields and cropland are presented in the next section.

The contribution of bioenergy to the total energy supply increases generally from the reference case to the mitigation case (except SSP5 "Taking the highway"), but is especially pronounced for the mitigation scenario of SSP2 "Middle of the road" and SSP4 "A road divided" (Fig. 3). Consequently, the global cropland area for bioenergy production increases rapidly in the



mitigation scenarios; as much as ten times for SSP4 "A road divided" between 2000 and 2100 (Fig. 4, panel c). The rapid expansion of cropland area for bioenergy production is the main driver for increases in total global cropland area for the mitigation scenarios of SSP2 "Middle of the road" and SSP4 "A road divided" (Fig. 4, panel d). Even for SSP1 "Taking the green road" bioenergy production is the prevailing driver of cropland expansion. However, in this scenario the cropland

expansion for bioenergy is counteracted by the sustainable lifestyle choices (e.g., decreasing meat consumption) and strong increases in yields, which together lead to a reduction of cropland area for food production. Quite differently, very low levels of technological change and thus very slow yield development paired with a strongly increasing population (12.1 billion people in 2100) result in the massive expansion of global total cropland for SSP3 "A rocky road" in both the reference and mitigation scenario. The trend of expanding and stabilizing global cropland in most scenarios is contrasted by the

development of global cropland for SSP5 "Taking the highway" in which global cropland increases and peaks in the first half of the 21st century, declining in the second half. The initial cropland expansion is due to the resource-intensive lifestyle of a slightly growing, much richer population, while bioenergy production does not play a role in this fossil fuelled scenario. For SSP5 "Taking the highway" strong yield developments and a declining population with saturated food demands lead to global cropland contraction in the second half of the 21st century.

**3.2 Impact of mitigation on consumption, yields and cropland area**

The avoided damage to GWP due to the introduction of mitigation strategies is at almost 8% in 2100 largest for SSP4 "A road divided", followed by SSP2 "Middle of the road" and SSP3 "A rocky road" (each around 5% in 2100, Fig. 5). The consumption of milk and meat is dependent on income and thus higher income associated with lower climate-induced damage enables additional consumption. For SSP3 "A road divided" the additional global average meat consumption (due to

avoided damage) is close to 1 kg meat per capita in 2100 (Fig. 5). In developing countries additional meat consumption is up to 1.5 kg meat per capita in 2100. However, compared with uncertainties within the relationship of income and meat consumption, as well as in lifestyle and cultural preference (e.g., for SSP3 "A rocky road" the global average meat consumption in 2100 is $52 \pm 10$ (1 SD) kg meat per capita (Engström et al., 2016a), such an impact of mitigation on per capita meat consumption appears relatively modest.

Differently to meat and milk (similar impact as for meat consumption, not shown) consumption, global average yield and global cropland area can be affected not only by avoided damage to GWP, but also by changed bioenergy production and yield. Each SSP's yield is the result of weighing the yield simulated by the ecosystem model under each RCP depending on the distance of the SSP's concentration pathway to that RCP (see Sect. 2.2.3 and Appendix A5). For example, for the yield of SSP2 "Middle of the road" the yields from RCP2.6, RCP4.5, RCP6.0 and RCP8.5 were weighted with 0.09, 0.15, 0.63

and 0.12 respectively (these numbers are called the "yield distribution", see Appendix A5 for yield distributions of the other scenarios). The yield distributions change when mitigation strategies are introduced, as the concentration pathway for each SSP is changed (Fig. 4, panel a). Due to investments in agricultural management, yield development is assumed also to be dependent on income, and thus avoided damage to GWP can increase yields in mitigation scenarios relative to reference





scenarios. For all scenarios, the avoided damage to GWP leads to slightly larger global average yields in the mitigation scenario compared to the reference case (up to 1%, Fig. 6, panel a). Changes in yield distributions have a stronger positive impact in the mitigation scenario of SSP1 "Taking the green road" (almost 3%) and SSP2 "The middle of the road" (1.5%), while the impact on yield in SSP5 "Taking the highway" is negative. Increased bioenergy production in the mitigation

scenarios has a very small impact on global average yield, as this impact is only indirect due to different allocation of cropland areas (areas with lower or higher yields). By contrast, the increased bioenergy production is the absolute largest contributor to the difference in global cropland area between mitigation and reference scenarios (Fig. 6, panel b). This is most strongly so for SSP2 "Middle of the road" and SSP4 "A road divided". Factors that resulted in larger yields in mitigation scenarios (avoided damage and yield distribution) counteract the cropland expansion caused by the increased

bioenergy production (higher yields, less cropland expansion), though with only a few percentage points when compared to the magnitude of the direct impact of bioenergy on cropland expansion in mitigation scenarios.

**3.3 Spatially explicit cropland changes and impact on the terrestrial carbon balance**

Cropland expansion in 2100 compared to 2000 (Fig. 7) can be observed in all scenarios in Sub-Saharan Africa, Brazil, Mexico, and in the Corn Belt and the Great Plains of North America. In the reference scenarios (all except SSP3 "A rocky

road") and also in the mitigation scenarios of SSP1 "Taking the green road" and SSP5 "Taking the highway" this cropland expansion is paired with cropland abandonment (green areas in Fig. 7) in other parts of North and South America, as well as in Eastern Europe and to some extent in Asia and Australia. An exception to this general pattern is SSP3 "A rocky road", where cropland expansion is predominant across all fertile lands globally. This is due to the combination of high population growth with resource intensive lifestyles as well as low yield increases. In the mitigation scenario of SSP3 "A rocky road",

bioenergy is mainly produced from crops grown in Brazil and the US (150 Mha each), but also Russia and Indonesia (50 Mha each). Even in other mitigation scenarios with large increases in cropland for bioenergy production (> 600 Mha in SSP2 and SSP4 compared to reference scenario), the increase is mainly allocated in Brazil, the US, Russia, Indonesia and to a lesser extent in India, Canada and Australia. The same pattern of cropland allocation for bioenergy production can be observed for the mitigation scenario of SSP1 "Taking the green road" (200 Mha more cropland for bioenergy production

compared to reference scenario). Interestingly, the very similar global aggregated cropland areas of the mitigation scenarios of SSP1 "Taking the green road" and SSP5 "Taking the highway" in 2100 (1725 Mha and 1722 Mha respectively) are distributed differently in the two scenarios: in SSP5 "Taking the highway" cropland changes led to more concentrated cropland areas in e.g. Sub Saharan Africa, Central America and Russia, while in SSP1 "Taking the green road" there are more subtle changes over larger areas, e.g. in Brazil, the US, Indonesia, but also Sub-Saharan Africa.

The large expansions of cropland areas in SSP3 "A rocky road" causes widespread carbon losses in 2100 compared to 2000, with up to -50 kg m$^{-2}$ in the tropics (Fig. 8). Even in temperate zones where cropland expands into previously forested areas, larger carbon losses occur. Also in scenarios with comparatively modest cropland expansion compared to SSP3, terrestrial carbon stocks decrease, especially in tropical regions and regions with cropland expansion (Fig. 7).




Climate change leading to a longer growing season in temperature-limited high latitude ecosystems, and increases in ecosystem productivity caused by the direct effect of rising $CO_2$ on the biochemistry of photosynthesis, have been identified as important drivers of the carbon balance of the terrestrial land surface (Le Quéré et al., 2015; Schimel et al., 2015), as simulated here for high latitude regions in all scenarios, and for wet tropical regions such as the Amazon and Congo Basin in all scenarios except to some extent SSP3.

After aggregating the changes in the terrestrial carbon pool at the global scale we found that the terrestrial biosphere is a carbon sink for most scenarios throughout the 21$^{st}$ century, but becomes a carbon source for scenarios with large cropland expansion (SSP3 "A rocky road" and SSP4m "An unequal world", Fig. 9). The global net-increase for most scenarios is not necessarily primarily driven by LULCC, but by the effects of climate change and $CO_2$ fertilisation (as described above). To isolate the effect of climate change vs. LULCC we performed simulations with constant land-use but changing climate (see Appendix A8, Fig. A1). These simulations suggest that without LULCC, the global terrestrial biosphere would act as a carbon sink for all scenarios (Appendix A8, Fig. A1). This would be most strongly the case for scenarios predominantly driven by high concentration pathways (RCP6.0 and RCP8.5, arriving at approximately 2175 GtC in 2100), but even for scenarios driven by the low concentration pathway RCP2.6 (arriving at approximately 2115 GtC in 2100).

LULCC erodes terrestrial C stocks for all scenarios by around 50 to 200 GtC by 2100. For SSP3 "A rocky road" the effect of the large-scale cropland expansion outweighs the climate change-driven sequestration of terrestrial carbon and the terrestrial biosphere turns into a net carbon source in the second half of the 21$^{st}$ century. This occurs more strongly for SSP3m compared to SSP3r, mainly due to lower assumed atmospheric $CO_2$ concentrations in the mitigation scenario (higher weighting of low radiative forcing RCP scenarios; Appendix A5), resulting in less $CO_2$ fertilisation of plant production, an affect expressed particularly in the simulated carbon balance of the tropics (Figs. 8,9). Production of bioenergy for mitigation and the related increase in cropland area (> 800 Mha cropland for bioenergy production in 2100) contributes to shifting affected areas from a carbon sink into a carbon source, as seen for SSP4m "An unequal road" (Fig. 9). In the fossil fuelled SSP5 "Taking the highway", bioenergy production does not increase, but the global carbon tax still reduces energy demand through enhanced energy efficiency, resulting in lower emissions, reflected in a greater weighting towards RCP6 in SSP5m and towards RCP8.5 in SSP5r (Appendix 5). However, the combined effects of climate, atmospheric $CO_2$ and land use result in almost identical carbon trajectories for the reference and mitigation cases of SSP5 (Fig. 9). SSP1m "Taking the green road" is the only scenario with expansion of cropland for production of bioenergy where the biosphere continues to be a strong carbon sink through the 21$^{st}$ century (Fig. 9).



## 4 Discussion

### 4.1 Findings in the context of other studies

We present a novel IAM framework and provide consistent SSP-scenario quantifications for energy supply, atmospheric carbon concentration, climate-induced damage to GWP and bioenergy production from energy crops, exploring impacts on
food consumption, cropland change and terrestrial carbon storage. So far, the literature only includes scenario quantifications for a subset of SSPs, and for a limited set of scenario factors. For example, in the IPCC Fifth Assessment report, three models project future reference primary energy use in 2100 to range from 1350 to 1850 EJ in 2100 (Bruckner et al., 2014), which is a slightly narrower range than the reference primary energy that emerges from our analysis (1030-2150 EJ in 2100). Preliminary SSP-scenario quantifications are available from the SSP database and suggest that primary energy for the
complete set of all five SSPs will range from 667 to 1920 EJ in 2100 for the reference case, and from 466 to 1363 EJ in 2100 under mitigation [considering only the marker scenarios and choosing the mitigation scenario with the RCP that is closest to our realization, (SSP-Database, 2015)]. For the mitigation scenarios, this compares to 1087-1252 EJ of primary energy in 2100, estimated by the climate economy model in our study. In the SSP quantifications, energy from biomass production increases for all reference scenarios (differently to our projections even for SSP5 "Taking the highway"), and is much more
pronounced in the mitigation scenarios. For example, for SSP4 "A road divided" the mitigation scenario simulated with GCAM4 (SSP4-26-SPA4-V12) projects primary energy use from biomass of 392 EJ in 2100, compared to 124 EJ in 2100 in the reference scenario (SSP4-Ref-SPA0-V12). This increase is similar as in our study (reference: 191 EJ in 2100 vs. mitigation: 519 EJ in 2100) and is also accompanied by a strong increase in cropland area due to mitigation (2812 Mha and 1722 Mha in 2100 in the mitigation and reference case, respectively; 2777 Mha and 1962 Mha, respectively, in our study)
(SSP-Database, 2015). Cropland projections for the preliminary SSP-quantifications cover a range from 1433 to 2812 Mha, which is comparable to the ranges previously published in the literature. For example, Schmitz et al. (2014) analysed cropland development until 2050 for scenarios based on SSP2 "Middle of the road" and SSP3 "A rocky road", taking into account a range of climate projections and agro-economic models, arriving at a range from 1400 Mha to 2300 Mha in 2050. A later model inter-comparison study (Alexander et al., under review) including a larger set of models and scenarios
(including all five SSPs) arrived at global cropland projections of 1100 to 2700 Mha in 2100. Our cropland projections for all scenarios, except SSP3 "A rocky road", are within the range reported by other studies and modelling teams (1546 to 2777 Mha in 2100). Cropland projections for SSP3 "A rocky road" extend beyond this range (3950 Mha and 4237 Mha in reference and mitigation respectively); reasons are discussed below. Biomass losses in conjunction with the extreme cropland increases projected under this scenario provide the major explanation for terrestrial ecosystems becoming a carbon
source in our analysis.

Since pre-industrial times, LULCC has contributed 180 ± 80 GtC or about one-third of total anthropogenic $CO_2$ emissions, to the atmosphere (Ciais et al., 2013). Biomass loss in conjunction with tropical deforestation is an important source, but is



compensated in part by a sink due to forest regeneration on abandoned cropland, e.g. in conjunction with agricultural intensification in mid-latitude countries after World War II (Shevliakova et al., 2009). If LULCC effects on biosphere carbon balance are disregarded, a residual carbon sink averaging $3.0 \pm 0.8$ GtC yr$^{-1}$ for 2005-2014 (Le Quéré et al., 2015) reduces the increase in atmospheric greenhouse gas concentrations due to anthropogenic emissions by around one quarter. Some

60% of this biospheric sink has been attributed to $CO_2$ fertilization (Schimel et al., 2015), while most of the remainder may be explained by a temperature-driven increase in growing season length in higher latitudes, enhancing vegetation productivity and creating a temporary sink for carbon in the stems of growing trees (Ahlström et al. 2012). For the future, our simulations suggest that for scenarios with wide-spread cropland expansion and slow agricultural intensification (SSP3 "A rocky road") biomass loss could turn the terrestrial biosphere once again into a carbon source. LULCC has been

previously shown to influence the carbon balance simulated by LPJ-GUESS, resulting in a general increase in carbon flux to the atmosphere under cropland expansion (Pugh et al., 2015). In all scenarios except SSP3 the residual carbon sink outweighs LULCC-induced carbon loss and the terrestrial biosphere sequesters $1.1 \pm 0.4$ GtC yr$^{-1}$ for 2000-2100 (or $1.9 \pm 0.3$ GtC yr$^{-1}$ for 2000-2100 disregarding LULCC). An earlier scenario study (based on the earlier, SRES scenario framework) suggested an average net sink of 2-6 GtC yr$^{-1}$ for 1990-2100 but in contrast to our scenarios, three of the four

underlying scenarios assumed a decrease in cropland areas (Levy et al., 2004). More recent estimates for the period 2000-2009 suggested a terrestrial carbon sink of $1.1 \pm 0.1$ GtC yr$^{-1}$ (Houghton et al., 2012), which is in the same range as our results for the 2000-2100 period.

The introduction of a global carbon tax as a mitigation strategy paired with the socio-economic characteristics of the SSPs results in varying reductions in atmospheric carbon concentrations in the range spanned by RCP2.6 and RCP6.0. Scenarios

with low challenges for mitigation (SSP1 "Taking the green road"), especially when combined with high challenges for adaptation (SSP4 "A road divided") achieve mitigation pathways that are comparable to the stringent mitigation scenario RCP2.6. However, as the imposed carbon tax only applies for fossil fuels, indirect emissions of land-use change caused by increased bioenergy production in the mitigation scenarios are not considered in these emission reductions. For SSP4 "A road divided", the terrestrial biosphere becomes a source of carbon in the second half of the 21st century and makes it

unlikely that SSP4 "A road divided" truly achieves a concentration pathway comparable to RCP2.6. By contrast, if socio-economic conditions – such as environmentally-conscious life-style choices (low-meat diet) paired with low population increase and strong technological growth – enable the reduction of cropland needed for food production, bioenergy from energy crops can be produced on the abandoned food-cropland and the biosphere as a whole acts as a sink, as for SSP1 "Taking the green road". This supports previous studies (Erb et al., 2012; Haberl et al., 2011; Kraxner et al., 2013) that point

out that only under specific conditions is bioenergy production sustainable and can contribute to mitigation of climate change.

In context with the mitigation strategies it is remarkable that the introduction of only 10% of the optimal carbon tax leads to significant energy, and thus atmospheric carbon concentration, reductions (e.g. SSP3 "A rocky road", 35% energy reduction



and 26% atmospheric carbon concentration reduction compared to reference scenario in 2100). Thus, if high damages are expected (as in SSP3 "A rocky road") even the introduction of a carbon tax that is far from optimal is a surprisingly effective strategy to mitigate climate change. However, this is under the assumption that the global carbon tax is introduced immediately and no further delays in climate change mitigation occur. Due to inertia in the climate system, the early

reduction of GHG emissions is crucial for the long-term effectiveness of any mitigation strategy (Luderer et al., 2013), but this is a very large challenge for the global community. Thus, especially for scenarios with high challenges for mitigation, the reductions in atmospheric carbon due to reduced fossil fuel consumption suggested by our study are on the optimistic side of available estimates. In comparison, the SPAs that accompany the SSP marker scenarios assume specifically different lengths of transition phases until full global climate cooperation is reached (and transition towards a globally uniform carbon

price thereafter), where the most ambitious SPA assumes complete transition by 2020 and is only used for SSPs with low challenges for mitigation (Riahi et al., 2015). A second key assumption in the SPAs concerns the extent of land-based mitigation. For examples, for SSPs with high affluence and high equality (SSP1 "Taking the green road" and SSP5 "Taking the highway") it is assumed that all land use emissions are taxed with the same level of carbon prices as in the energy sector (Riahi et al., 2015). In our study, the mitigation scenario for SSP5 "Taking the highway" achieves a concentration pathway

just below RCP6. To reach a more stringent RCP, such as RCP2.6, land-based mitigation options would need to be considered, such as afforestation projects or carbon capture and storage (see Sect. 4.3 for further discussion). Excluding emissions from land-use in mitigation strategies has previously been observed to lead to large scale land-use change (Wise et al., 2009), as simulated here for SSP2 "Middle of the road" and SSP4 "A road divided".

**4.2 Uncertainties in the IAM framework and input data**

The presented scenario outcomes should be treated as illustrative, as a wide range of outcomes can arise  due to uncertainties in interpretations and quantification of scenario assumptions (Engström et al., 2016a). For example, the uncertainty range for one scenario of cropland change was 1330-1750 Mha by 2100 from 1500 Mha in 2000 (1SD, Engström et al., 2016a). Differences in model structure can likewise cause large spread in results. For example, using one scenario, but 10 different models, Schmitz et al. (2014) projected cropland changes ranging from 1400-2000 Mha by 2050 (relative to a baseline of

1500 Mha in 2000), depending on the model chosen (despite harmonized input data). Differences were related to diverse model assumptions as to land availability, costs for land conversion and endogenous yield responses to technological change (Schmitz et al., 2014). Similarly, the future fate of the net biospheric sink for carbon is highly uncertain, with biospheric models projecting divergent trajectories in net carbon balance depending on the increase in atmospheric $CO_2$ associated with a given emissions scenario (e.g. RCP), the climate patterns and trends projected by different GCMs in response to the

emissions, differences in ecosystem response simulated by different biosphere models, and whether or not biogeochemical and biophysical biosphere-atmosphere feedbacks are taken into account (Ahlström et al., 2012; Boysen et al., 2014; Cramer et al., 2001; Friedlingstein et al., 2014; Sitch et al., 2008). The IPSL climate model chosen to provide climatic forcing for the ecosystem model simulations in our study induces carbon cycle changes in the middle of the range of an ensemble forced by


multiple GCMs (Ahlström et al., 2012, Fig. A 2 in Appendix A.8). Another example of the importance of model structure is the simulated cropland change for SSP3 "A rocky road" of 3950 Mha in reference scenario, compared to an earlier quantification of SSP3 "A rocky road" with a mean of 2280 Mha with identical parameter settings, but an earlier version of the land-use model used here (Engström et al., 2016a). The structural changes in the updated model version are related to the

intensification of the livestock sectors, the trade mechanism, and bioenergy production. Previously, the trade mechanism allowed substantial underproduction, which was assumed to be avoided in the updated model version. Thus, the increased demand (11% higher cereal demand due to allowed intensification of the livestock sector and 15.2 EJ bioenergy from crops in 2100 in SSP3 "A rocky road"), paired with the fulfilment of demand, leads to the very high cropland projections for SSP3 "A rocky road". The cropland expansion in SSP3 is further driven by very low global average yield increase (3.2 ton ha$^{-1}$ in

2100 and 3.0 ton ha$^{-1}$ in 2000, compared to 5.2 ton ha$^{-1}$ in 2100 for SSP5 "Taking the highway"), which is partly also caused by the damage to GWP and thus reduced investments in agricultural technologies.

Additionally to uncertainty arising from model structure, different interpretations of scenarios as well as the translation of qualitative scenario information into quantitative scenario parameterizations contribute to uncertainties of scenario outcomes. For the land-use model used here, the effect of uncertainties in input parameters was shown to produce scenario

outcomes with uncertainty ranges ($2 \times SD$) of up to 27% of the scenario mean (Engström et al., 2016a). For the outcome of the climate-economy model uncertainties arise especially due to parameter interdependencies of substitution elasticity with increase in growth rates for fossil fuels and clean energy. Slightly different parameter combinations can lead to very different outcomes. Other parameters, such as the discount rate, were not varied here, but likewise have the potential to change model outcomes (Golosov et al., 2014).

However, the relative similarity of our results to the preliminary SSP quantifications (SSP-Database, 2015) gives us confidence that our projections are plausible both in the direction and magnitude of change and can serve as examples for the quantification of the SSPs based on a coherent logic.

### 4.3 Limitation of the study and further research

One limitation of the presented study already alluded to is the restriction of the mitigation strategies to the energy sector. It

would be a valuable addition to the modelling framework to include other land based mitigation strategies, e.g. avoided deforestation and afforestation, as well as demand-side mitigation strategies (meat-low diets, reduction of food waste) which have been previously shown to have a great mitigation potential (Smith et al., 2013). For bioenergy production it would be desirable to introduce other crops that are shown to have greater mitigation potential, such as C$_4$ grasses and switch-grass (Albanito et al., 2016). Additionally, the information flow within the IAM framework could be improved. One such

improvement would be to inform the climate-economy model about land-use based emissions due to bioenergy production derived from the ecosystem model. The assumption that clean energy (of which bioenergy is one part) is free from emissions should then be revised or bioenergy modelled as a separate energy source. Further research could also improve the treatment of the uncertainties outlined above. A systematic sensitivity study of the climate-economy model paired with a sensitivity



study performed earlier for the land-use model (Engström et al., 2016a) would give better insights into the impact of energy mitigation strategies on consumption, yield and cropland. Also, the impact of cropland changes and mitigation-derived reduced climate change on the terrestrial carbon balance could be quantified with an ensemble of GCMs to account for the uncertainties in climate forcing arising from structural differences among GCMs.

**5 Conclusions**

Our results suggest that the indirect impacts of climate mitigation strategies on global cropland are small in comparison to impacts due to the spread of bioenergy production and other sources of uncertainties, such as model structure and uncertainties in parameterizations. We found that different drivers, such as food production vs. bioenergy production, can lead to contrasting land-use change patterns, as observed here for SSP1 "Taking the green road" and SSP5 "Taking the
highway". Further, without substantial increases in global average crop yields, feeding the global population of 12.1 billion in 2100 assumed under SSP3 "A rocky road", additionally to producing bioenergy, will cause serious deforestation and transforms the global terrestrial carbon pool to a sink of carbon emissions. Our study thus underlines previous assertions that bioenergy production from energy crops is only a sustainable mitigation strategy if other socio-economic factors, such as population growth, technological change and lifestyle choices, free up existing cropland areas for allocation to bioenergy
production.

**Acknowledgments**

We acknowledge funding from the Formas Strong Research Environment grant "Land use today and tomorrow" (LUsTT; dnr: 211-2009-1682) and the Swedish Research Programme for Climate, Impacts and Adaptation (SWECIA) funded by the Foundation for Strategic Environmental Research (Mistra). Peter Alexander is thanked for supply of initial values of
bioenergy production, calculated from FAOSTAT (2015). This study is a contribution to the Strategic Research Area Biodiversity and Ecosystem Services in a Changing Climate (BECC).

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

## 5 Appendix

### A1. PLUM development

In the earlier published version of the land use model (Engström et al, 2016a) it was assumed that the feed ratio would be constant at year 2000 levels, resulting in potential underestimation of the feed demand for scenarios where meat and milk consumption increase strongly, given the assumption that with increased consumption of animal products the production
intensifies, leading to a higher demand for cereal feed. In the PLUM version used here it is assumed that feed ratio increases proportionally to the increase in consumption of animal production (half the growth), but maximal to a scenario-specific feed ratio maximum (*5_feedRatioCap*). Country level data of the feed ratio and per capita meat consumption from the year 2000 (FAOSTAT, 2016), show that only very few countries have feed ratios that exceed 0.4 (independent from per capita meat consumption). For countries with feed ratios of 0.4 or lower the data suggest a weak correlation ($R^2$=0.14) with per capita
meat consumption. As this relationship is quite weak, we made a conservative assumption and chose values of 0.1, 0.2, 0.1, 0.2 and 0.3 for the 5_feeedRatioCap scenario parameter for the SSP1-5 respectively.

The estimates of potential arable land were updated using the areas classified as moderate to very high suitability for high input level rain-fed cereals (Suitability and Potential Yield) by the Agro-ecological Zones Data Portal (FAO/IIASA, 2011), compared to the previous PLUM version (Engström et al, 2016a).

### 20 A2. Downscaling GWP

The damage to GWP was distributed among countries based on their share of Gross Domestic Product (GDP) on GWP. In this case the burden of climate-change induced damage to GWP is divided equally. However, to ensure consistency with the social equity assumption in the scenarios, two alternative approaches were implemented. First, for scenarios with high social equity we assumed that high income countries (HICs) would pay a larger share, i.e. 80%, of the damage, while 20% would
be paid by middle income countries (MICs) and none by low income countries (LICs). Second and vice versa, in scenarios with limited social equity low income countries would need to pay 50% of costs, middle income 40% and high income only 10% of costs. The reasoning in the second alternative is that low income countries are the most vulnerable to the impacts of climate change, but would not receive much support by high income countries to deal with the consequences of climate change. Categorization of countries into high, middle and low income countries is based on the baseline year 2000 and
countries in these income groups contribute with 63%, 29% and 8% respectively to GWP in 2000 (Table A. 1, Baseline).



**Table A. 1.  Percentages of GWP of high, medium and low income countries for reference scenarios without downscaled damage (woD) and with downscaled damage (wD) for baseline year 2000 and in 2100.**

| Scenario (equity) | HIC | | MIC | | LIC | |
|---|---|---|---|---|---|---|
| | woD | wD | woD | wD | woD | wD |
| Baseline | 63 | 63 | 29 | 29 | 8 | 8 |
| SSP1 (high) | 23 | 21 | 35 | 36 | 41 | 43 |
| SSP2 (medium) | 22 | 22 | 38 | 38 | 40 | 40 |
| SSP3 (low) | 20 | 21 | 45 | 46 | 35 | 33 |
| SSP4 (medium) | 33 | 33 | 41 | 41 | 26 | 26 |
| SSP5 (high) | 30 | 26 | 32 | 33 | 38 | 41 |

Throughout the 21st century for all scenarios the share of HICs decreases and shares of MIC and LIC countries increases (Table A. 1). Our assumptions for the distribution of damage reinforce this pattern for scenarios with high equity, while the pattern is weakened in scenarios with low equity, see Table A. 1.

**A3. Clean energy to bioenergy**

Clean energy projections (nuclear and renewable energies) from the climate economy model were disaggregated to receive bioenergy projections by a) converting energy from climate economy model from Gtoe into EJ, multiply with 41.868 (1 Gtoe = 41.868 EJ); b) calculating nuclear energy by multiplying the total energy from the climate economy model (oil, coal, clean energy) with the share of nuclear energy from the WEO scenarios (see Table A. 2); and c) calculating bioenergy by subtracting nuclear energy from clean energy (= total renewable energies) and multiplying with the share of bioenergy on total renewable energies (bioenergy and other renewable energies).

**Table A. 2.  Shares of energy sources on total primary energy demand in 2010 and in 2035 for the WEO scenarios (OECD/IEA, 2012).**

| Share on primary energy (%) | 2010 | current policy | new policy | 450ppm |
|---|---|---|---|---|
| Fossil fuels | 81 | 80 | 76 | 63 |
| Nuclear energy | 6 | 6 | 7 | 11 |
| Bioenergy | 10 | 9 | 11 | 15 |
| Other renewable energies | 3 | 5 | 6 | 12 |





### A4. Bioenergy in PLUM, technical documentation

The following steps were implemented to include bioenergy from energy crops ($BE_{cr}$) in PLUM:

The bioenergy from the climate economy model ($BE_t$ in EJ) is used as input to PLUM (after interpolating to get annual values, assuming a starting value of 45 EJ in 2000, first value of climate economy model is in 2010 and varies around 50-60

EJ). The share of bioenergy that is produced from agricultural energy crops ($worldBEcr_t$ in EJ) is calculated with (Eq. A1), where $shareBEcr_i$ (unitless) is the share of bioenergy from energy crops on total bioenergy, which was 3% in 2008 (OECD/IEA, 2012). The remaining bioenergy from the climate economy model is produced using other feedstock: 67% of fuel wood, 20% of forest residues, 4% of agricultural by-products, 3% of animal by-products and 3% of waste in 2008 (OECD/IEA, 2012). Additionally the scenario parameter $shareBEcr_{13}$ accounts for that this share might increase in the

future, if for example traditional bioenergy from wood fuel is replaced with modern bioenergy from energy crops. Estimates are that up to 25-30% of total bioenergy could be produced from energy crops by 2050, and the highest value for $shareBEcr_{13}$ is 30%, in PLUM achieved by 2100 (time()=0-100, for the years 2000-2100). In $shareBEcr13$ "13" indicates that this is scenario parameter number 13.

$$worldBEcr_t = BE_t * shareBEcr_i + BE_t * \left( \left( \frac{shareBEcr_{13}}{100} \right) - shareBEcr_i \right) * \frac{time()}{100} \qquad \text{(Eq. A1)}$$

$worldBEcr_t$ is the net primary bioenergy from energy crops and does include energy that is lost during the conversion of biomass to bioenergy. The energy content of the biomass produced for bioenergy from energy crops in 2000 was 2.09 EJ (calculated using crop specific energy contents), while primary energy supply was estimated to 1.35 EJ. This implies an initial conversion efficiency of 1.55 (=2.09EJ/1.35EJ), $conversionEff_i$. It is assumed that the conversion efficiency ($conversionEff_t$) can be improved over time (max 1.55-0.5=1.05) and the scenario parameter $efficiencyBEcrEJ_{14}$ is introduced

(0-50%), see Eq. 2.

$$conversionEff_t = conversionEff_i - \frac{14\_efficiencyBEcr}{100} * \frac{time()}{100} \qquad \text{(Eq. 2)}$$

The global energy content in biomass from energy crops ($worldBEcr_tgross$, EJ) is then calculated as in Eq. 3.

$$worldBEcr_tgross = worldBEcr_t * conversionEff_t \qquad \text{(Eq. 3)}$$

$worldBEcr_tgross$ needs to be distributed to the 160 countries (n=1-160) in PLUM, see Eq. 4. This is done with help of

$cF\_BEtotal_{i,n}$ (unitless), the per country fraction of bioenergy on total bioenergy production, as well as $yieldBEcr_{i,n}$, the country specific yield of bioenergy ($yieldBEcr_{i,n}$, EJ/Mha). FAOSTAT commodity balance sheets, "other uses" for crops cereals total, vegetable oils, and sugar crops (Mt) and production and production area were used to derive initial bioenergy area and yields (Alexander et al, 2015; FAOSTAT, 2015). The category "other uses" covers bioenergy, as well as materials and stimulants. For the selected crops it was assumed that all of "other uses" is used for bioenergy and only commodities that

are not agricultural by-products were selected to insure consistency with the WEO definition of energy crops "Energy crops – those grown specifically for energy purposes, including sugar and starch feedstocks for ethanol (corn, sugarcane and sugar





beet), vegetable-oil feedstocks for biodiesel (rapeseed, soybean and oil palm fruit) and lignocellulosic material (switchgrass, poplar and miscanthus)" (OECD/IEA, 2012). Lignocellulosic material was excluded here.

The resulting per country bioenergy cropland demand $croplandBEcrD_{t,n}$ has the unit 1000 ha ((EJ/EJ*Mha)*10^3=1000ha).

$$croplandBEcrD_{t,n} = \frac{worldBEcr_tbrutto * cF\_BEtotal_{i,n}}{yieldBEcr_{i,n} * yieldGrowth_{t,n}} * 10^3 \qquad \text{(Eq. 4)}$$

In equation Eq. 4 it assumed that the yield of bioenergy changes proportional to the change (mostly growth) in yield, $yieldGrowth_{t,n}$ (unitless) as simulated for cereal yield ($cYield_{t,n}$ and $cYield_{i,n}$), see Eq. 5.

$$yieldGrowth_{t,n} = 1 + \frac{cYield_{t,n} - cYield_{i,n}}{cYield_{i,n}} \qquad \text{(Eq. 5)}$$

The change in cropland demand for bioenergy production per country is calculated in Eq. 6.

$$\Delta croplandBEcrD_{t,n} = croplandBEcrD_{t,n} - croplandBEcrD_{t-1,n} + extraCroplandBEcrD_{t,index} \qquad \text{(Eq. 6)}$$

There is an extra demand for cropland for bioenergy ($extraCroplandBEcrD_t$, in 1000 ha), due to the fact that some countries approach their maximum of arable land and cannot produce the bioenergy demanded from them. This extra demand is divided among countries that have more than three times the minimum residual naturally vegetated potential cropland available and have cropland for bioenergy in the first simulation year ($resNV\_L$, in 1000 ha), see Eq. 7. $worldCroplandBEcrD_t$, $worldCroplandBEcr_t$ and $worldResNV\_L$ are global sums of $croplandBEcrD_{t,n}$, $croplandBEcr_{t,n}$ and

$resNV\_L$ respectively.

$$extraCroplandBEcrD_t = \frac{resNV\_L}{worldResNV\_L} * (worldCroplandBEcrD_t - worldCroplandBEcr_t) * 1000 \qquad \text{(Eq. 7)}$$

Cropland area for bioenergy production ($croplandBEcr_{t,n}$ in 1000 ha) is initialized with $croplandBEcr_{i,n}$ (in 1000 ha) calculated by using the FAOSTAT data referred to above. Changes in cropland for bioenergy are taken/given from/to forest and grassland, see Eq. 8.

$$croplandBEcr_{t,n} = croplandBEcr_{i,n} + forestCroplBEcr_{t,n} + grasslandCroplBEcr_{t,n} \quad \text{(Eq. 8)}$$

For $forestCroplBEcr_{t,n}$ (1000 ha) and $grasslandCroplBEcr_{t,n}$ (1000 ha) rules similar as for the conversion of forest and grassland to cropland are applied as previously described for PLUM development in Engström et al., (2016). These rules describe that a scenario dependent share of total land is always reserved for natural vegetation (defined in $x_{t,n}$ and $y_{t,n}$), see Eq. 9 and Eq. 10. For land conversion for bioenergy the share of land reserve for natural vegetation is assumed to be double

the amount reserved under conversion process for food production, in order to prioritize food production.

$$forestCroplBEcr_{t,n} = \Delta croplandBEcr_{t,n}D * x_{t,n} * y_{t,n} \qquad \text{(Eq. 9)}$$

$$grasslandCroplBEcr_{t,n} = \Delta croplandBEcr_{t,n}D * (1 - x_{t,n}) * y_{t,n} \qquad \text{(Eq. 10)}$$

Cereal production for "other uses" is otherwise included through the overproduction demand in PLUM ($overPro$, $overPro_i$ = 30% initial value in 2000) and this initial overproduction demand needs to be reduced by the production of cereals for "other

uses" which is now included explicitly for bioenergy production. To adjust the initial overproduction demand the share of cereal production for bioenergy ($proShareBEc_i$, unitless) on the $worldDemand_i$ (Mt) in 2000 is calculated, see Eq. 11;

$$proShareBEc_i = \frac{\sum_{n=1}^{160} cProBEc_i}{worldDemand_i} \qquad \text{(Eq. 11)}$$





where the cereal production for bioenergy $cProBEc_i$ (in Mt) is calculated taken from FAOSTAT data for the year 2000. Finally, $overPro$ is adjusted, see Eq. 12, where $overProd_1$ is a scenario parameter that adjust overproduction demand over time (Engström et al., 2016).

$$overPro = (overPro_i - proShareBEc_i) + (overPro_i - proShareBEc_i) * overProd_1 * time() \qquad \text{(Eq. 12)}$$

5 **A5. SSP-RCP matrices**

To derive the values for the SSP-RCP matrices we calculated for each scenario (SSPr1-5, SSPm1-5) the distance from the simulated changes in atmospheric carbon pool (Fig. 4, panel a) to all RCPs in 2100 (as the RCPs are defined by their targets in 2100). The normalised distance indicates how likely a given SSP will result in a given RCP, i.e., the smaller the distance, the higher the probability that the SSP will result in the RCP modelled by the climate economy model. The inverse of the

10 distances were normalised, resulting in the probabilities in Table A.3.

For the weighing of LPJ-GUESS NECB, only RCPs with probabilities above 0.1, or if all four RCPs have probabilities above 0.1 then the three highest (as in SSP5 reference), were included.

**Table A. 3. Matrices for reference (r) and mitigation (m) scenarios with probabilities that a given SSP results in a given RCP.**

|      | RCP 2.6 | | RCP 4.5 | | RCP 6 | | RCP 8.5 | |
|------|-------|-------|-------|-------|-------|-------|-------|-------|
|      | r | m | r | m | r | m | r | m |
| SSP1 | 0.018 | 0.760 | 0.961 | 0.148 | 0.016 | 0.063 | 0.005 | 0.029 |
| SSP2 | 0.092 | 0.482 | 0.155 | 0.348 | 0.633 | 0.119 | 0.120 | 0.051 |
| SSP3 | 0.035 | 0.248 | 0.065 | 0.582 | 0.864 | 0.123 | 0.036 | 0.047 |
| SSP4 | 0.024 | 0.918 | 0.047 | 0.047 | 0.906 | 0.023 | 0.022 | 0.011 |
| SSP5 | 0.112 | 0.074 | 0.156 | 0.155 | 0.282 | 0.712 | 0.450 | 0.058 |



**A6. Rationale for the parameter settings of the climate economy model for reference and mitigation scenarios**

In reference SSP1 energy technologies are directed away from fossil fuels due to a low growth rate in the efficiency of coal extraction (0.5% annually in the time period 2010-2100 compared to 2% baseline). Simultaneously the efficiency of renewable energy production is allowed to grow at 2.5% annually over the time period 2010-2100. Energy efficiency is

regarded as an integral part of the low carbon and low energy intense future of SSP1. Energy efficiency is stimulated by relatively high energy prices (implemented via a lower substitution elasticity of 0.80). The societies are well prepared for possible effects of climate change (low damage factor of $5 \times 10^{-5}$). Due to the overall importance of sustainable and holistic solutions, the challenge to mitigation is low. Mitigation strategies in the SSP1 mitigation scenario include an optimal carbon tax, as well as measures to reduce the growth rate of the efficiency of coal extraction to 0%. Measures to facilitate green

energy substitution are implemented increasing the substitution elasticity to 0.95.

In reference SSP2 the continued reliance on fossil fuel keeps the growth of extraction efficiency for coal at a moderate pace (2% annually). Some investments in renewable energy technologies leads to growth of renewable energy efficiency at 1.5% annually. In SSP2 reference there are no significant improvements in energy infrastructure or energy efficiency projects, thus substitution elasticity remains at 0.85 throughout the period 2010-2100. Medium challenges to adaptation are parameterised

with a damage factor of $10 \times 10^{-5}$. With a moderate challenge to mitigation, mitigation strategies for SSP2 include an increase in growth of efficiency of renewable energies to 2% annually, while the growth in extraction efficiency of remains unchanged (2.0% annually). Due to the mitigation strategies there are slight improvements in infrastructures for renewables energies (substitution elasticity 1.05), as well as global carbon tax that covers 30% of the social costs of carbon.

In SSP3 reference scenario slow technological change mostly directed towards domestic energy sources leads to a continued,

albeit slower growth of efficiency of coal extraction (1.2% annually) and a moderate growth of renewable energy production efficiency (1.0% annually), as some regions growingly rely on domestically available renewable energy sources. Energy and carbon intensities remain high and the substitutability of different energy sources is at a medium level. The mitigation challenges for SSP3 are very high, but only limited mitigation efforts are undertaken. The growth rate of renewable energy production efficiency to 1.2 % annually, while the substitution elasticity remains at 09.95. Additionally in the SSP3

mitigation scenario a very moderate (10% of optimal) global carbon tax is introduced. However, SSP3 also has a high challenge for adaptation and the damage factor is high $15 \times 10^{-5}$, leading to high expected damages and additional incentives to decrease energy use under the mitigation scenario.

In reference SSP4, the substitution elasticity is slightly below the reference and the growth rates of efficiency coal and renewable energy are moderate, both at 1.5% annually. Challenges to mitigation are high, characterized by a high damage

factor ($15 \times 10^{-5}$), resulting in large anticipated damage from fossil fuel burning and thus overall lower fossil fuel use in the mitigation scenario. In the SSP4 mitigation scenario additional efforts are made to increase production of renewable energies, both through increasing the production efficiency growth of renewable energy (to 2% annually), reducing the





growth rate of coal extraction to 1% annually and improving energy infrastructure investments (a substitution elasticity of 1.05). As the challenges to mitigation are low, an optimal global carbon tax is implemented under the SSP4 mitigation scenario.

In reference SSP5 all energy development is focused on fossil fuels, leading to very high growth rates for extraction

efficiency of coal, 2.2% annually and growth in renewable energy production efficiency is neglected (0%). The price is a massive use of fossil fuel at cheap prices and low energy efficiency. Fossil fuels are predominantly used for all of energy production facilitated by a high substitution elasticity of 1.05. However, as the increased level of development lowers the challenge to adaptation, the damage of expected climate change is buffered by a low damage factor ($5 \times 10^{-5}$). Nevertheless, challenges to mitigation are high but the mitigation strategies for SSP5 include only a very moderate global carbon tax (10%

of the optimal), a higher substitution elasticity (1.2) and a slight decrease in growth of coal extraction efficiency (2%) compared to the SSP5 reference scenario.

**A7. Details of cropland downscaling**

Country level changes in land cover were downscaled to $0.5 \times 0.5°$ grid cells using a suitability index ($S_{i,t}$) where the suitability for cropland in grid cell $i$ was calculated based on the proximity of cropland in surrounding grid cells and the

potential crop productivity in the target grid cell. If the projected country level change in cropland cover was positive ($\Delta LC_t > 0$), the following algorithm was used:

$$S_{t,i} = \frac{\sum_{n=1}^{N} \left(\frac{LC_{n,t-1}}{d_n^\gamma}\right)^\alpha + \left(\frac{LC_{i,t-1}}{d_i^\gamma}\right)^\alpha}{\sum_{n=1}^{N} \left(\frac{1}{d_n^\gamma}\right)^\alpha + \left(\frac{1}{d_i^\gamma}\right)^\alpha} P_{i,t-1}^\beta$$

and for a decrease ($LC_t < 0$):

$$S_{t,i} = \frac{\sum_{n=1}^{N} \left(\frac{1 - LC_{n,t-1}}{d_n^\gamma}\right)^\alpha + \left(\frac{1 - LC_{i,t-1}}{d_i^\gamma}\right)^\alpha}{\sum_{n=1}^{N} \left(\frac{1}{d_n^\gamma}\right)^\alpha + \left(\frac{1}{d_i^\gamma}\right)^\alpha} P_{i,t-1}^\beta$$

where:

$N$ is the number of neighbors of grid cell $i$ (8 except in coastal areas);

$d_n$ is the distance to grid cell $n$;

$LC$ is cropland cover;

$\alpha$ is the scalar of the distance factor, could differ between increasing or decreasing $LC$;

$\beta$ is the scalar of the production factor, could differ between increasing or decreasing $LC$;

$\gamma$ is a distance weighting factor;

$P$ is potential crop productivity, simulated by the ecosystem model.





In this study, values used were α: 1.0/1.0 (increase/decrease), β: 2.0/1.0 (increase/decrease), γ: 2.0.

The resultant suitability index was then divided by the number of grid cells in the target country to yield the grid cell share of the country-level land cover change:

$$\Delta LC_{i,t} = \frac{S_i}{\sum_{h=1}^{H} S_h} \Delta LC_t$$

$$LC_{i,t} = LC_{i,t-1} + \Delta LC_{i,t}$$

5    where:

$H$ is the total number of cells in a country.

**A8. Additional figures**

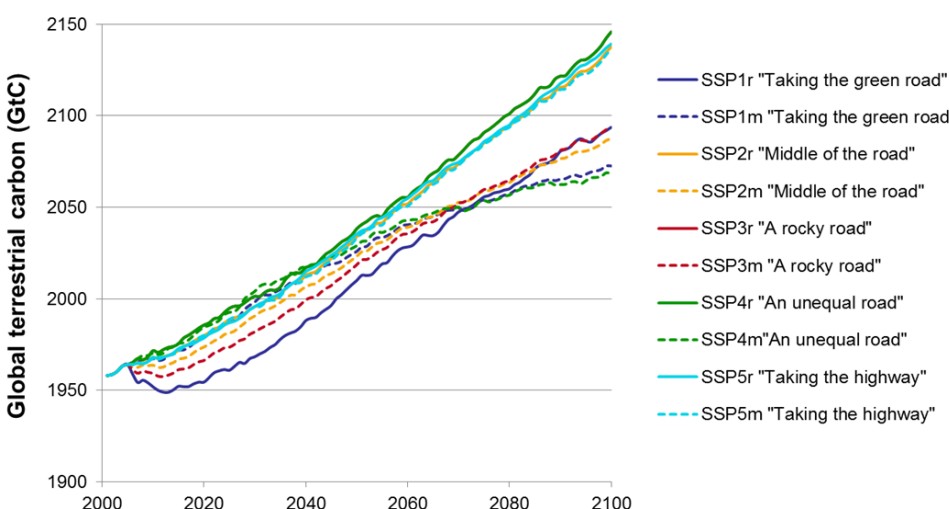

10    **Figure A 1: Global terrestrial biosphere carbon (GtC) simulated with constant year 2000 land cover forced by climate change fields from the IPSL GCM. Scenarios with large shares of RCP8.5 and RCP6.0 (SSP2r, SSP3r, SSP4r, SSP5r and SSP5m) achieve highest global terrestrial carbon in 2100, scenario mainly driven by mainly RCP2.6 (SSP1m and SSP4) result in the lowest global terrestrial carbon, while scenarios mainly driven by RCP4.5 (SSP1r, SSP2m and SSP3m) place in between.**





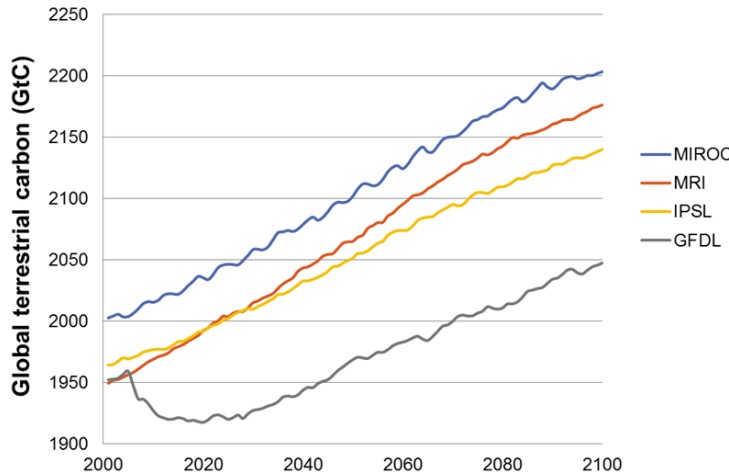

**Figure A 2: Global terrestrial biosphere carbon (GtC) simulated with climate forcing fields from four GCMs (MIROC (Watanabe et al. 2010), MRI (Yukimoto et al. 2012), IPSL (Dufresne et al., 2013) and GFDL (Donner et al., 2011) for medium LULCC (SSP2m) and RCP4.5 driven climate change. Uncertainties due to different process representations in the four GCMs arrive at**
5    **differences across the GCMs in 2100 which are similar to the total simulated change for one GCM (e.g. difference MIROC-GFDL is around 150 GtC in 2100, while total simulated change of GFDL is around 100 GtC or 200 for MIROC). The initial dip in GFDL terrestrial carbon is mainly caused by Amazon forest dieback.**





**Figures**

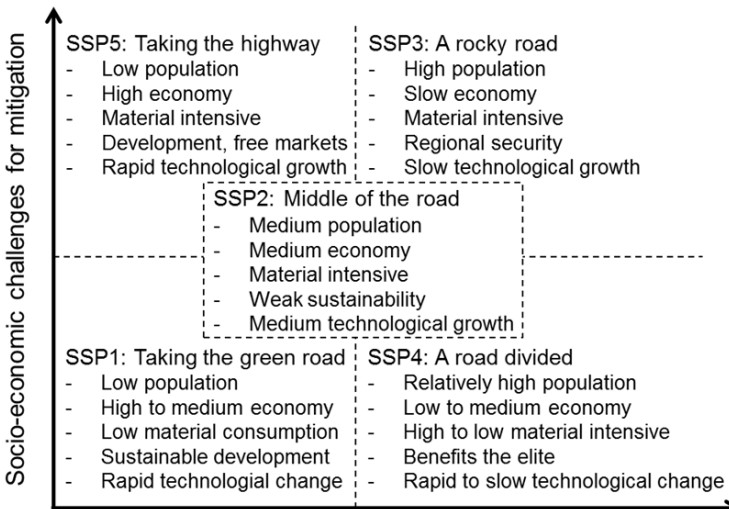

**Figure 1. The SSP scenarios of global socio-economic development in the 21st century in their space of challenges for mitigation and adaption (adapted from O'Neill et al. 2013) and their development in selected key elements: growth of population, growth of the economy, lifestyle, policy orientation and technological development (O'Neill et al., in press).**





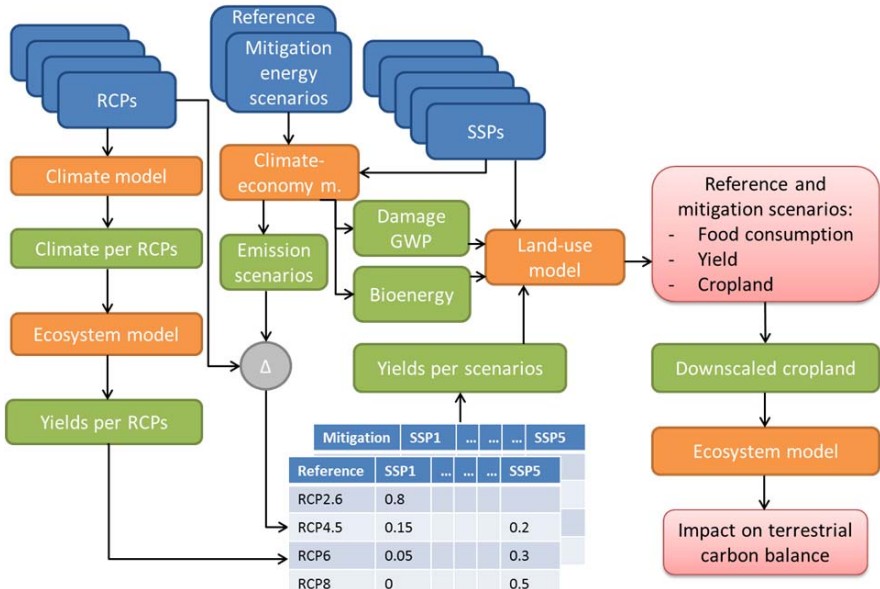

**Figure 2. Overview of the Integrated Assessment Modelling (IAM) framework showing input data sets in blue, component models in orange and information flows/intermediate results in green. Final results are displayed in red. The Representative Concentration Pathways (RCPs) are input to the climate model and the Shared Socio-economic Pathways (SSPs) are input to the**
5  **climate-economy model and the land-use model. Damage to gross world product (GWP) is input to the land-use model. Δ signifies the distances between emissions predicted by the climate economy model and implied by RCPs, used as inverse weights the create yield time series as input to the land use model.**





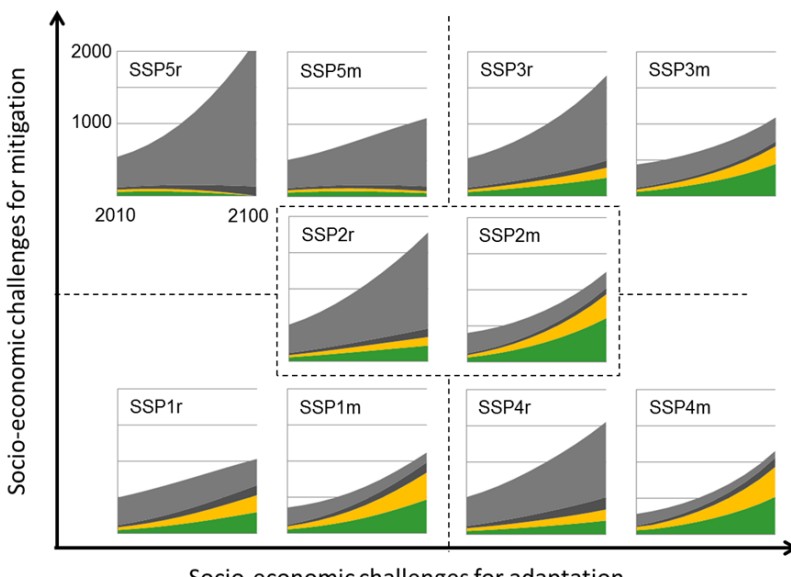

**Figure 3. Primary energy demand (0-2000 EJ; vertical axis of internal panels) predicted by the climate-economy model for the reference (r) and mitigation (m) versions of each SSP scenario (see Fig. 1) for the time period 2010-2100 (horizontal axis of internal panels).**



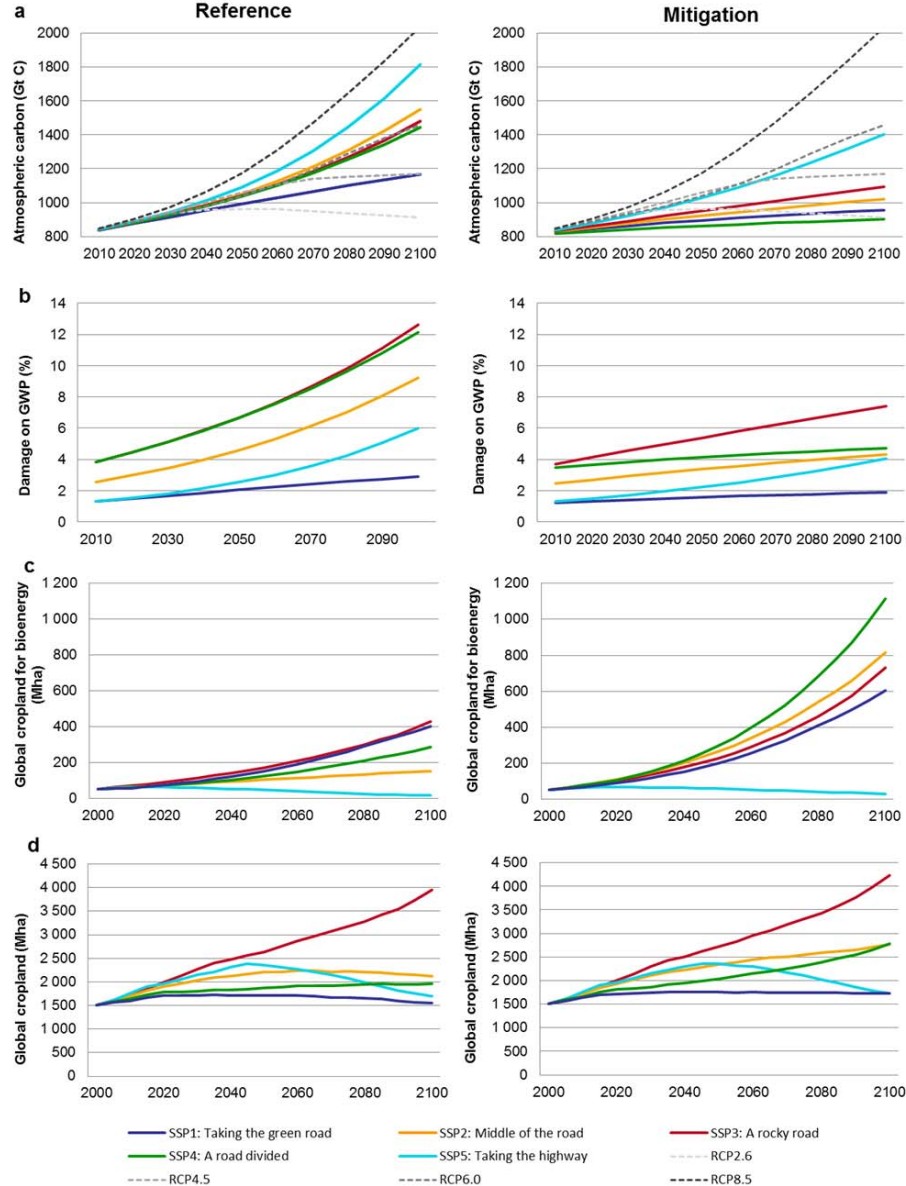

**Figure 4. (a) Atmospheric carbon (GtC) including carbon trajectories (GtC) for the four RCPs, (b) corresponding damage to GWP (%), (c) global cropland for bioenergy (Mha) and (d) global cropland (Mha) for reference and mitigation scenarios.**



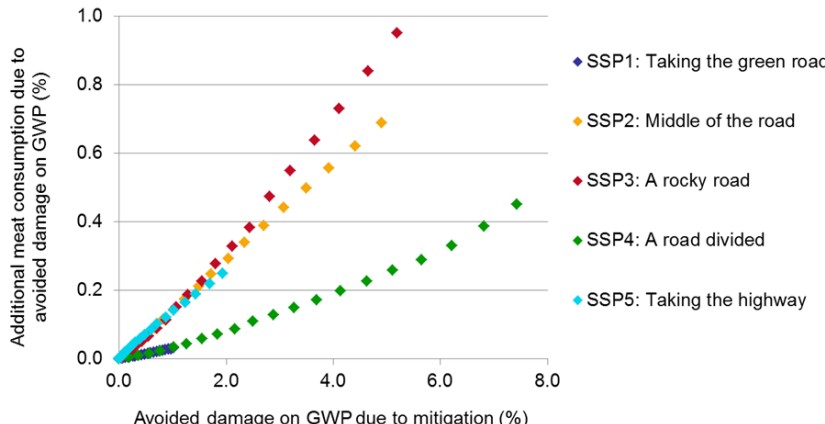

**Figure 5. Impact of avoided damage to GWP (% of total GWP) due to mitigation on global aggregated meat consumption in 2100 relative to 2000 for the five SSPs.**

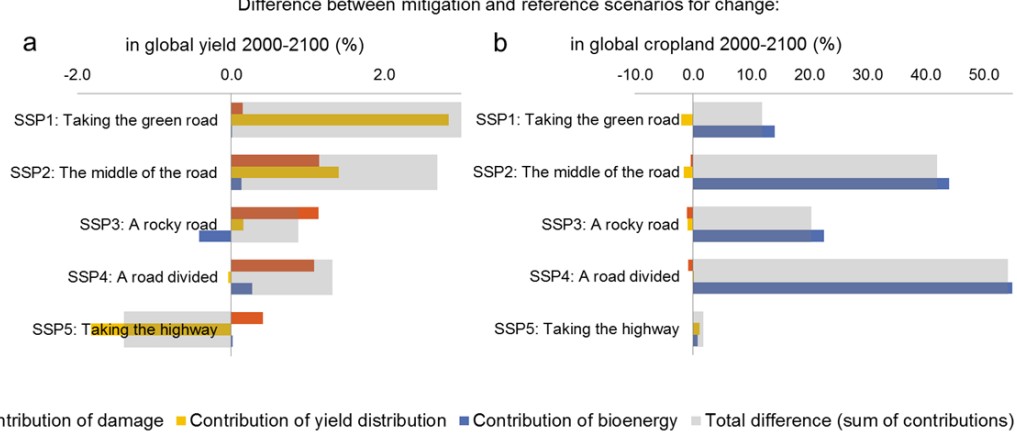

**Figure 6. (a) Difference between reference and mitigation scenarios for change in global yield between 2000 and 2100 (%) and (b) change in global cropland area between 2000 and 2100 (%). The grey bar gives the total differences (sum of differences due to damage, yield distribution and bioenergy), while the colored bars show the contribution of each causal factor to the total difference.**



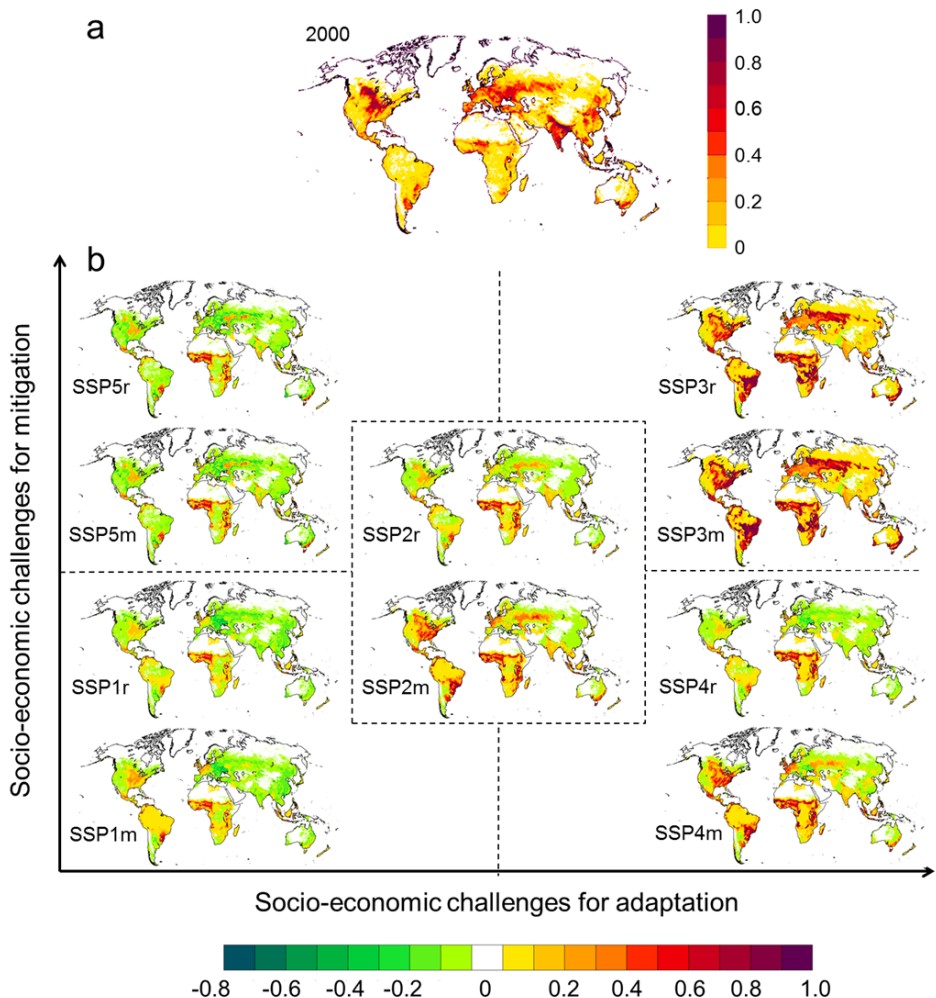

**Figure 7. (a) Fraction of cropland on total land area for baseline year 2000 based on Hurrt et al. (2011). (b) Simulated cropland changes relative to baseline by 2100 for the five SSP reference (r) and mitigation (m) scenarios in the challenge for adaptation and mitigation space. Green colours indicate a decrease in cropland area in 2100 compared to 2000, while yellow and red colours indicate an increase in cropland area in 2100 compared to 2000.**



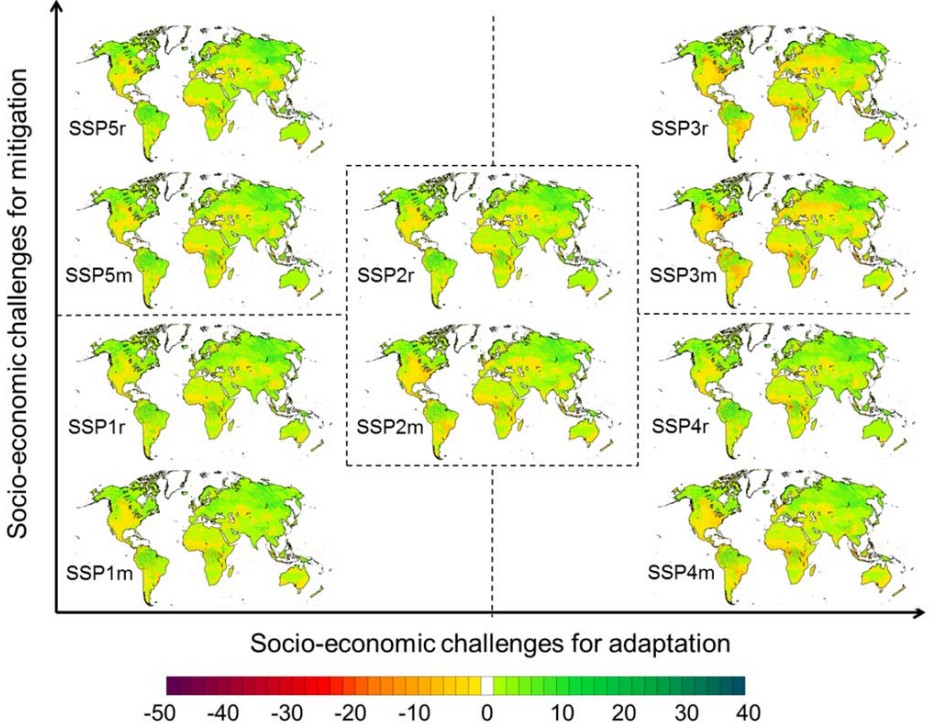

**Figure 8. Changes in modelled total terrestrial carbon pool (kg m$^{-2}$) for 2000-2100. Green to blue colours indicate an increase in the carbon pool, while yellow and purple colours indicate a decrease in carbon pool.**



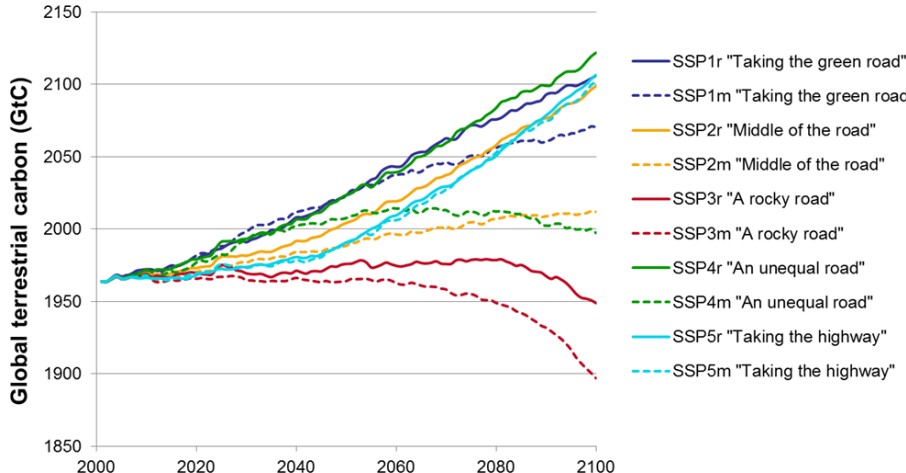

**Figure 9: Changes in global terrestrial biosphere carbon pool (GtC) from 2000 to 2100 for all SSPs, reference (r) and mitigation (m) case. The slope indicates whether the net ecosystem carbon balance (NECB) is a carbon source (negative slope) or carbon sink (positive slope).**

## 5    Tables

**Tabl1. Parameters in climate economy model modified for scenarios.**

| Parameter | Abbreviation | Baseline value | Unit |
|---|---|---|---|
| Growth in production efficiency of coal | $A_{2,g}$ | 2 | annual growth in % |
| Growth in the efficiency of clean energy technologies | $A_{3,g}$ | 2 | annual growth in % |
| Substitutions elasticity between different energy sources | $se$ | 0.95 | |
| Damage elasticity factor | $\gamma$ | 2.38 | $10^{-5}$ per airborne GtC |
| Level of carbon tax | $\tau$ | 0, 1 | fraction of optimal carbon tax |



**Table 2. Challenges for mitigation and adaptation and energy-related key elements for the five SSPs.**

| Key element | SSP1: Taking the green road | SSP2: The middle of the road | SSP3: A rocky road | SSP4: A road divided | SSP5: Taking the highway |
|---|---|---|---|---|---|
| Challenge for adaptation | low | medium | high | high | low |
| Challenge for mitigation | low | medium | high | low | high |
| Energy technological change | Directed away from fossil fuels, toward efficiency and renewables | Some investment in renewables but continued reliance on fossil fuels | Slow technological change, directed toward domestic energy sources | Diversified investments including efficiency and low-carbon sources | Directed toward fossil fuels; alternative sources not actively pursued |
| Carbon intensity | low | medium | high in regions with large domestic fossil fuel resources | low/medium | high |
| Energy intensity | Low | Uneven, higher in LICs | High | Low/medium | High |
| Fossil constraints | Preferences shift away from fossil fuels | No reluctance to use unconventional resources | Unconventional resources for domestic supply | Anticipation of constraints drives up prices with high volatility | None |



**Table 3. Parameter settings in the climate economy model (see Table 1) for reference (r) and mitigation (m) scenarios based on the SSPs and the challenge for adaptation (damage elasticity factor, $\gamma$) and mitigation (carbon tax, $\tau$, as a proportion of optimum, for mitigation scenarios).**

| Scenario | $A_{2,g}$ | | $A_{3,g}$ | | $se$ | | $\gamma$ | | $\tau$ | |
| --- | --- | --- | --- | --- | --- | --- | --- | --- | --- | --- |
| | r | m | r | m | r | m | r | m | r | m |
| SSP1 "Taking the green road" | 0.5 | 0.0 | 2.5 | 2.5 | 0.80 | 0.95 | 5 | 5 | 0.0 | 1.0 |
| SSP2 "The middle of the road" | 2.0 | 2.0 | 1.5 | 2.0 | 0.85 | 1.05 | 10 | 10 | 0.0 | 0.3 |
| SSP3 "A rocky road" | 1.2 | 1.2 | 1.0 | 1.2 | 0.95 | 0.95 | 15 | 15 | 0.0 | 0.1 |
| SSP4 "A road divided" | 1.5 | 1.0 | 1.5 | 2.0 | 0.90 | 1.05 | 15 | 15 | 0.0 | 1.0 |
| SSP5 "Taking the highway" | 2.2 | 2.0 | 0.0 | 0. 0 | 1.05 | 1.20 | 5 | 5 | 0.0 | 0.1 |





**Table 4. Parameter settings in the PLUM land use model for *feedRatioCap* (0.1-0.3: feed ratio increases for countries with feed ratios below 0.1-0.3 up to 0.1-0.3, that is a maximum 10%-30% of meat is produced with cereal feed for countries with initially low feedRatios), *equity* (1=high equity distribution, 0=equal distribution, -1=low equity distribution), the share of bioenergy that is produced from bioenergy crops in 2100 (*shareBEcr*, %, 3% being the initial value in 2100), the conversion efficiency of energy in biomass to bioenergy that is achieved in 2100 (*efficiencyBE*, %, 64% being the initial value in 2000) for reference (r) and mitigation (m) scenarios based on the SSPs.**

| Scenario | feedRatioCap | | equity | | shareBEcr | | efficiencyBE | |
|---|---|---|---|---|---|---|---|---|
| | r | m | r | m | r | m | r | m |
| SSP1 "Taking the green road" | 0.1 | 0.1 | 1 | 1 | 6 | 6 | 68 | 70 |
| SSP2 "The middle of the road" | 0.2 | 0.2 | 0 | 0 | 3 | 6 | 66 | 68 |
| SSP3 "A rocky road" | 0.1 | 0.1 | -1 | -1 | 6 | 6 | 64 | 66 |
| SSP4 "A road divided" | 0.2 | 0.2 | 0 | 0 | 6 | 9 | 66 | 68 |
| SSP5 "Taking the highway" | 0.3 | 0.3 | 1 | 1 | 3 | 3 | 64 | 64 |