# Peer review of "Impacts of climate mitigation strategies in the energy sector on global land use and carbon balance"

_Earth System Dynamics, 2016_

## Referee Comment (RC1) · P. Kyle (Referee) · 31 Oct 2016

—Summary—

The authors have pulled together a lot of material to address the research questions, which are quite complex and required the development of several new modeling tools that haven't had any prior publications. I would want to see changes to the modeling processes prior to recommending publication, and some more references to the literature on applying climate damages in an IA modeling context.

—Several points—

* My first comment pertains to the energy/economy model used here: I don't see any

value to publishing a new set of SSP-ish scenarios from this very simple model that appears to be parameterized inconsistently with the corresponding publicly available SSP scenarios. The sophistication of the energy/climate model in this study is similar to, or less than, the IA models in the 1980s. This wouldn't be a problem as long as the simple model were parameterized so as to replicate the results of the larger energy-economy models used to produce the SSPs (in similar fashion to the simple climate models that replicate the results of the GCMs). Most of the parameters that this model takes as exogenous inputs are the product of complicated and generally non-linear dynamics, and instead of just being guessed (e.g., technology efficiency improves at 2%/yr from 2010 to 2100), they should be calculated from those more detailed models' outputs. Much more effort should be focused on validating that the outputs from the energy/economy and land model here are in fact consistent with the published scenarios. That comparison should be done for all key variables assessed here in order to ensure consistency; the discussion includes mostly anecdotal observations that two of the ten scenarios here have similar cropland quantities and total primary energy demands as two of the scenarios in the SSP database.

* I also have a problem with the basic design of the study, but this is really a decision for a journal editor and not a reviewer, and there's not really anything that could be done to change it. The study uses a detailed crop and vegetation model to represent climate impacts at the 0.5 x 0.5 degree scale, but then uses an extremely simple multiplier on a nation's GDP (or the world's GWP) to calculate the climate damages. I am aware that others in this field do that, and so perhaps there is no issue here. But in my opinion, climate impact-related damages simply do not lend themselves well to that sort of simplistic representation.

Climate impacts, by their nature, are non-linear with respect to global temperature, variable over time, region-specific, and context-dependent. In the form of droughts and extreme events, they are also relevant at sub-annual time scales, below the resolution of the timesteps being represented in the global energy/economy/land models. To esti-

mate the costs of climate impacts in any region and time period, one would first need to know what the physical climate impacts are; second what the direct damages are; and third what the adaptive capacity of the system is, along with the costs of adaptation. At this point, the scientific community has produced scenarios of climate at the appropriate temporal and spatial scales, and is currently working on how to model the impacts of the climate on the relevant activities in the economic, energy, and agricultural sectors. This study doesn't address the complexities of climate impacts in estimating GWP losses; it uses a simple "marginal damage" function that relates economic productivity loss to the $CO_2$ concentration. I know they cited another study that used/developed that function, but in my opinion there is no reason to believe that this relationship has any validity, applied to a future economy that is likely very different from today's, and with climate impacts that include much more than temperature change. Given the current state of the art in the impacts, adaptation, and vulnerability (IAV) community, I doubt this relationship was demonstrated to hold for a variety of nations with different climate impacts and different economic structures.

* The authors should specify what the downscaled gross world product (GWP), to the country level, is used for. The method is documented in the text and appendix, but I never saw what subsequent calculations it was used for; it may be used to modify a country's GDP and therefore energy demand, but I'm not sure. I don't particularly like the method, as it doesn't consider the inter-national differences in climate impacts; for instance, temperature increase could be good for economic productivity in some countries (e.g., Sweden) while bad in others (e.g., India). Also it doesn't consider that climate impacts will affect different sectors of the economy in different fashion (e.g., agriculture vs manufacturing vs services vs household), so that the climate impacts on GDP will be different for countries with different economic structures (all else equal).

* More documentation of how the climate impacts were applied to the agricultural sectors should be provided. In this sort of vegetation and agro-economic model link-up, many countries typically see unrealistic and positive yield impacts, particularly places

with a harsh climate and low yields in the historical years, where small increases in precipitation can lead to large modeled yield increases. In my work with similar data I've had the most trouble with the Middle East, North Africa, Russia, and Canada. But to some extent this depends on the mathematical formulations for applying aggregated crop model output to the baseline nation-level yield trajectories.

* Next, I'll address a few of the simplifications and representations that struck me as particularly problematic in the modeling exercise; unfortunately, without the raw data inputs and outputs to/from the model, I can really only guess as to the relative importance of each.

1) shareBEcr: this parameter, exogenous in all periods and scenarios, represents the combustible energy content of all ethanol and biodiesel feedstocks divided by total global bioenergy demands. The denominator includes all remaining uses of bioenergy, which the authors note account for some 97% of the base-year bioenergy demands. The basic problem is that these bioenergy commodities (in the denominator) have fundamentally different future demand drivers from ethanol and biodiesel (the numerator), so there isn't really any way to know a priori how this will evolve over time, in the various narratives of the SSPs.

In the current study design, the authors are attempting to set the "shareBEcr" such that the quantity of agricultural crops used as bioenergy feedstocks does not grow by more than 30-50% from its base year value, according to the estimates of a study (Haberl et al. 2010). However, in the model, this is applied as a share constraint rather than a quantity constraint, so the target quantity (from Haberl et al. 2010) appears to be greatly exceeded in some if not all of the scenarios. On the other side, the bioenergy commodities that do grow a lot (up to 450 EJ/yr) are the unspecified ones, which in the study methods are not tied to cropland or the land/carbon models, even though it is stated that this commodity class includes ligno-cellulosic (i.e., "second-generation") bioenergy sources. These bioenergy crops are a very important component of future land use change in the SSP scenarios, and probably account for the vast majority of

the growth of bioenergy here. This is because traditional uses of firewood and charcoal, and industrial recycling of bio-derived byproduct fuels, are simply not energy demands that are likely to scale up in any significant way in response to an emissions mitigation policy. So, by bunding second-generation bioenergy crops with waste and traditional biomass commodities whose production is not tied to land use, the scenarios are getting up to 450 EJ/yr of bioenergy, almost as high as total global primary energy consumption of all fuels today, without causing land use change or any other consquences relevant for emissions and carbon stocks.

2) conversionEff: this parameter describes the relationship between the combustible energy content of harvested bioenergy and the biofuels produced, in the form of ethanol and biodiesels. The authors estimate this efficiency at about 65% in the base year, with a maximum value (year 2100, with efficiencyBEcrEJ set to 50%) of 95%. The end-of-century levels are simply not realistic; that would entail conversion processes wherein the vast majority of the combustible energy content of the by-products (dried distillers grains and oil crop feedcakes) are somehow transferred into the fuel. I don't know what the theoretical limits on that conversion are, but I suspect it's closer to 65% than 95%.

3) A2 and A3: the annual improvement rate in the efficiency of producing coal and carbon-free energy, respectively. It is possible that this description is inaccurate in several ways; I'm hoping that what is intended is the improvement in the whole-economy energy intensity of the use of these fuels, or the ratio of primary (usable) energy to economic output. Improving the energy efficiency of producing these energy commodities (e.g., less fuel-intensive coal mining or farming practices) wouldn't make much difference to energy consumption at the global level, and in any case these practices are likely to become more energy-intensive over time, not less, due to resource depletion, mechanization of farming, and others. There are also problems if this were interpreted as the efficiency of using energy. An efficiency that grows at 2% per year from 2010 to 2100 ends up 6 times higher than it started, and for the maximum improvement rate

used, 2.5%/yr, it ends up nearly 10 times higher. There are no uses of coal in the energy system, at a global level, with thermal efficiency levels low enough to permit this sort of improvement.

And, like many parameters here, I would suggest calculating them from the model outputs in the publicly available SSP scenarios, and using some simplification from that calculation, rather than arbitrarily guessing. The SSP suggested parameterizations (guidelines) were written for IA models with a much higher level of detail of the physical systems than the tools used here.

4) Yield: the yield growth rates I would also suggest taking from the SSP database, using area-weighted and indexed cereal yields in each region. The current method assigns baseline productivity growth on the basis of the yield gap, from the Mueller et al gridded yield gap study. There are two issues with this approach. For one, as the authors note, the rate at which countries close the yield gaps is tied to "each scenario's technological growth, economic development and technology transfer." However, these attributes are more granular than the inputs to the model used, and it isn't specified how those yield trajectories were developed. Second, convergence with base-year yield gaps is only one component of future agricultural productivity improvements; the distribution itself should also shift upwards due to technological change. In regions with no or little yield gap (e.g., Europe, the USA), yield improvements to 2100 are effectively frozen in this method, which likely isn't what is intended.

5) p: the rate at which future welfare is discounted. Part of the problem with the research goals of this study is that the impacts of climate change from emissions today play out over hundreds of years, due to the long lifetime of CO2, not even taking into account issues like sea level rise or thermohaline cycle disruption. How the net present value of damages can be applied to an economy over such a long time span and across generations is a topic without consensus in the modern economic literature. Some review is warranted (e.g., Stern versus Nordhaus). Still, one point with good agreement is that the discount rate is very important for the balance between near-term emissions

mitigation and long-term reduction in climate damages. I couldn't find where the discount rate was stated, but did find a statement that the discount rate was not varied in any sensitivity analysis, so I'd suggest clarifying what is used, stating the justification, and running a couple of sensitivity scenarios.

—Specific items—

p2 line 10 - mitigation isn't solely for the purpose of decreasing negative impacts on human society. also for terrestrial biosphere (e.g., biodiversity, ecosystem function).

p4 lines 6-8: climate impacts isn't the only factor driving yield changes over time (also yield gap convergence) p4 line ∼20: how are energy supplies modeled, in order to get supplies and demands to balance? Are there exogenous supply curves used? p4 lines 20-21: all IA models represent energy markets explicitly, and have since the first-generation IA models back in the 1980's (e.g., Edmonds-Reilly-Barnes was first documented in 1986). p4 lines 23-25: given the complexities involved, I don't see how one can reasonably state that the % GDP loss is a linear function of the global average temperature, but given that it is another study that is being cited, please provide a 1-2 sentence description of how this was estimated in that study–over what time scale, geographic scale, temperature change, and was is an empirical estimate from historical data, or a model-derived estimate? It is crucially important for the results in this study, but strikes me as very questionable.

p5 line 2: the emissions pathways from this model should be compared with the published ssp's, and harmonized to the extent possible.

p5, lines 25-30: from my understanding of the methods later on, trade is set a priori and cropland expansion is used to modify the supply, so that it is equal to demand plus or minus net trade. this is a bit unusual in this field; in most models, trade is price-sensitive, and can be an important determinant of the equlibrium between agricultural production and demand. it woud be a good idea to make sure the results from this approach are reasonable in India, which has already very high cropland shares, and a

population that is growing fast and becoming more wealthy, both of which put significant upward pressure on agricultural product demands.

p6, line 29: it is stated that bioenergy is only produced on abandoned cropland; what is used to estimate abandoned cropland? I'm not aware of any inventories that dis-aggregate this quantity specifically, but there are vast quantities of land in the former Soviet Union (Central Asia), the Middle East, and the forests of the eastern United States that were cropland at some point in human history. it is hard to see how these lands would be the preferred sites for bioenergy production, particularly in light of the locations where cropland expansion is currently taking place (e.g., tropical rainforests).

p7 line 30: the hurtt et al (2011) dataset distinguished pasture on the basis of land use, not land cover class. it classified as pasture vast tracts of land area that are not grassland, including most of Tibet, Australia, Central Asia, and the western USA. it's probably not correct to assume this is all grass, but it might also not be important for the study; I can't tell. p8 line 10: irrigation, N application, and tillage intensity are held at base year levels while yield gaps are assumed to close. However, in Mueller et al. (2012), these were the main factors that account for present-day yield gaps.

p11 - for any grid cell, the yield impact is not a simple linear function of the radiative forcing. I'm not sure what is gained by using this probability-weighted approach as opposed to just simply assigning a single RCP scenario that is most similar to the emissions outcome of the given scenario.

Figure 4 - Please clarify whether global cropland (4d) includes global cropland for bioenergy (4c). It did in the SSP reporting database and in Schmitz et al. (2014), so hopefully it does here too!

---

## Referee Comment (RC2) · D.B. Kirk-Davidoff (Referee) · 9 Dec 2016

As editor, I am submitting a reviewer comment, thus closing the discussion, in light of the authors' long wait for the completion of this review process and of the first reviewer's excellent and thorough review.

The authors present a study of an integrated assessment model in which they first find parameter settings that allow the model to approximate a set of scenarios described in the Shared Socio-economic Pathways framework, and then add a mechanism intended to represent a carbon tax imposed on fossil fuel combustion, and note the impact of this tax on gross world product, on fossil fuel use, and on agricultural activity.

[Figure]

I concur with the first reviewer's comments and recommendations, especially with regard to the desirability of a more realistic accounting of the local cost to wealth of climate impacts (since these are already spatially resolved for use in the ecosystem model).

I have a number of additional suggestions for clarifications. It was not immediately clear to me that the various SSP scenarios would be imposed on the model through parameter choices (as opposed, for example to forcing the model through variable population growth rates or some other forcing mechanism). The introduction could be rewritten to make this much clearer, and also to address the first reviewer's concerns about how consistent the model trajectories are with the SSPs as defined.

The discussion of "damage on GWP" is confusing- this seems to be just a proxy for global warming averted, but since the climate-economy model calculates GWP explicitly, couldn't the GWP itself be shown, so that the increase in GWP due to the optimized carbon tax would be apparent in Figure 4? Similarly, the terms "challenges to adaptation" and "challenges to mitigation" don't seem to be as parallel in meaning as their grammatical parallelism would suggest. "Challenges to adaptation" seems to indicate political resistance to adaptation, while "challenges to adaptation" seems to indicate a structural likelihood of a lot of damage to GWP due to warming. Perhaps these should be rephrased as "resistance to mitigation" and "wealth available for adaptation" (which would have the opposite sign to "challenges to adaptation")? In this regard, l. 21 on page 8 seems problematic without a clear baseline: since higher $\gamma$ means more damage per unit Carbon emitted, the required reduction in emissions to achieve a given reduction in damage is actually less, though of course the reduction in emissions required to achieve a given low \*level\* of damage would be larger.

---

## Author Comment (AC1) · 3 Mar 2017

Response to P. Kyle (Referee)

Summary: The authors have pulled together a lot of material to address the research questions, which are quite complex and required the development of several new modeling tools that haven't had any prior publications. I would want to see changes to the modeling processes prior to recommending publication, and some more references to the literature on applying climate damages in an IA modeling context.

Response: Thank you for the thorough review of our study which has led to revisions that will surely help to improve the manuscript. The modelling tools we have pulled

together are all previously published (Climate economy model (Golosov et al., 2014), PLUM (Engström et al., 2016a; Engström et al., 2016b) and LPJ-GUESS (Lindeskog et al., 2013; Smith, 2001; Smith et al., 2014)), while the IAM framework linking them together is new. We believe that our modelling approach is defendable and holds value as an alternative method, compared with the IAM-generated "official" SSP projections, for interpreting the SSP scenarios and relating them to climate, emissions, ecosystem impact, land use and energy sector development in a coherent way. As noted below, there is no single "correct" way to interpret the SSP narratives, which describe different aspects of the possible future world in a qualitative and relative (to present, and to alternative futures) way. Thus there is value in the availability of multiple interpretation methodologies based on different but defendable assumptions and approaches, to stimulate debate and encapsulate uncertainty. Furthermore, our goal is not to make necessarily accurate predictions (given current uncertainty, how could this be judged?) but to investigate the plausible implications of key interactions between biophysical processes and human decision-making under climate change and its impacts.

Several points:

* My first comment pertains to the energy/economy model used here: I don't see any value to publishing a new set of SSP-ish scenarios from this very simple model that appears to be parameterized inconsistently with the corresponding publicly available SSP scenarios. The sophistication of the energy/climate model in this study is similar to, or less than, the IA models in the 1980s. This wouldn't be a problem as long as the simple model were parameterized so as to replicate the results of the larger energy-economy models used to produce the SSPs (in similar fashion to the simple climate models that replicate the results of the GCMs). Most of the parameters that this model takes as exogenous inputs are the product of complicated and generally non-linear dynamics, and instead of just being guessed (e.g., technology efficiency improves at 2%/yr from 2010 to 2100), they should be calculated from those more detailed models' outputs. Much more effort should be focused on validating that the outputs from the energy/economy

and land model here are in fact consistent with the published scenarios. That comparison should be done for all key variables assessed here in order to ensure consistency; the discussion includes mostly anecdotal observations that two of the ten scenarios here have similar cropland quantities and total primary energy demands as two of the scenarios in the SSP database.

Response: The general approach taken in this paper was to represent the global system governing emissions and land use with a parsimonious system. In contrast to other similar work, we used a modern macroeconomic model where the economy is represented as a set of markets where forward-looking agents makes decisions to maximize a well specified individual objective function – in other words, the model has microeconomic foundations. This contrasts to the approach taken in most existing IAMs used for the provision of RCP emission scenarios where either supply and demand are modelled ad-hoc or a welfare function for a representative household is maximized. Following the route we chose has both advantages and disadvantages. A disadvantage also true of state-of-the-art modern macro models is that they are very much less detailed in the description of, e.g., technical characteristics of energy supply. However, there are also advantages that made the macroeconomic community switch to models with microeconomic foundations decades ago. A key advantage is that by explicitly modelling the incentives and constraints of agents in the market, the models are in principle insensitive to the Lucas critique. This essence of this critique is that estimated historic correlations between aggregate variables cannot be trusted to be invariant to policy changes. Therefore, calibrating our model to simply replicate traditional and more detailed models, which is very reasonable in models of nature, is not as straightforward a way forward here. An advantage of modelling markets explicitly, rather than maximizing a welfare function like in RICE/DICE and MERGE, is that market imperfections, suboptimal tax policies as well as uncertainty can be introduced in a straightforward way. Relevant market imperfections to include in the analysis could be market power and asymmetric information. That there are both pros and cons of economic models based on micro-foundations make us argue that our approach is complementary to and

motivated in providing a relevant alternative approach to the more standard IAMs.

We are aware of that we in this article are far away from taking full advantage of the potential provided by models with microeconomic foundations. However, using a model that is very familiar to macroeconomists with training after, say the 1990s has the advantage that it can help bring more macroeconomists into climate-economy modelling. Focusing on essential processes describing the system requires assumptions for processes not explicitly modelled, in contrast to more sophisticated IAM frameworks that include the here excluded processes. With the parsimonious approach we aimed at providing an independent set of SSP realisations based on the SSP storylines and harmonized key input data, such as population and economic growth. The aim to provide consistent and independent SSP realisations and focusing of interaction effects is clarified in the introduction. The choice of parameter settings in the climate economy model was derived through the interpretation of SSP storylines and are clarified in the method section. Deriving growth rates from the SSP quantifications would in our view not be desirable, as this would compromise the independence of the developed scenarios. A comparison of the developed energy scenarios to published energy scenarios (as of winter 2015/2016, e.g. (Bruckner et al., 2014)) was done during the parameterisation. In the revised manuscript we added the comparison of the simulated energy scenarios with the SSP marker scenarios. In the introduction we clarified that the developed scenarios are not predictions but rather serve to highlight interactions that might be important, e.g. leading to non-obvious outcomes, in the coupled social-biophysical system.

* I also have a problem with the basic design of the study, but this is really a decision for a journal editor and not a reviewer, and there's not really anything that could be done to change it. The study uses a detailed crop and vegetation model to represent climate impacts at the 0.5 x 0.5 degree scale, but then uses an extremely simple multiplier on a nation's GDP (or the world's GWP) to calculate the climate damages. I am aware that others in this field do that, and so perhaps there is no issue here. But

in my opinion, climate impact-related damages simply do not lend themselves well to that sort of simplistic representation. Climate impacts, by their nature, are non-linear with respect to global temperature, variable over time, region-specific, and context-dependent. In the form of droughts and extreme events, they are also relevant at sub-annual time scales, below the resolution of the timesteps being represented in the global energy/economy/land models. To esti- mate the costs of climate impacts in any region and time period, one would first need to know what the physical climate impacts are; second what the direct damages are; and third what the adaptive capacity of the system is, along with the costs of adaptation. At this point, the scientific community has produced scenarios of climate at the appropriate temporal and spatial scales, and is currently working on how to model the impacts of the climate on the relevant activities in the economic, energy, and agricultural sectors. This study doesn't address the complexities of climate impacts in estimating GWP losses; it uses a simple "marginal damage" function that relates economic productivity loss to the $CO_2$ concentration. I know they cited another study that used/developed that function, but in my opinion there is no reason to believe that this relationship has any validity, applied to a future economy that is likely very different from today's, and with climate impacts that include much more than temperature change. Given the current state of the art in the impacts, adaptation, and vulnerability (IAV) community, I doubt this relationship was demonstrated to hold for a variety of nations with different climate impacts and different economic structures.

Response: Considering the complexity of vegetation-ecosystem processes, even the process-based vegetation model LPJ-GUESS can be considered as parsimonious. Despite intensive research efforts during the last decades the future evolution of the global carbon cycle is still the subject of considerable debate, with different state-of-the-art models yielding contrasting projections both in offline and coupled Earth system simulations (Ahlström et al., 2012; Friedlingstein et al., 2014). In model intercomparison studies, LPJ-GUESS typically exhibits mid-range responses compared to other models (e.g. Ahlström et al., 2012) and emerges as comparatively skillful in reproducing

ecosystem patterns and trends compared with independent estimates e.g. from satellites and flux towers (Murray-Tortarolo et al., 2013; Piao et al., 2013). It is one of the few globally benchmarked ecosystem models that accounts explicitly for demographic processes controlling the accumulation of carbon in growing forest stands following agricultural land abandonment – a significant component of the extant land carbon sink, and thus important to represent correctly (Shevliakova et al., 2009). There is no doubt that there is a very large amount of uncertainty about how climate change will affect the economy. We do not expect the uncertainty about economic effects far into the future and for large climate changes to fade anytime soon. We thus use the much used Nordhaus (2008) aggregate damage function that based on a large number of studies express economic damages as a function of the change in global mean temperature. Golosov et al. (2014) show that this, in combination with a logarithmic relation between atmospheric $CO_2$-concentation and temperature, produces a relation between $CO_2$ concentration and damages with a constant quasi-elasticity. To illustrate the great uncertainty, we provide scenarios, based on different parametric assumptions about future damages and technological development. This can provide a sense of orders of magnitude but should, obviously, not be taken as forecasts. The geographic distribution of damages could be done using more existing information. However, our results are not sensitive to this.

* The authors should specify what the downscaled gross world product (GWP), to the country level, is used for. The method is documented in the text and appendix, but I never saw what subsequent calculations it was used for; it may be used to modify a country's GDP and therefore energy demand, but I'm not sure. I don't particularly like the method, as it doesn't consider the inter-national differences in climate impacts; for instance, temperature increase could be good for economic productivity in some countries (e.g., Sweden) while bad in others (e.g., India). Also it doesn't consider that climate impacts will affect different sectors of the economy in different fashion (e.g., agriculture vs manufacturing vs services vs household), so that the climate impacts on GDP will be different for countries with different economic structures (all else equal).

Response: In section 2.1, we mentioned that the damage to GWP influences food consumption and yield development, but not energy demand, which is now elaborated in the revised manuscript. Generally, the use of spatially explicit climate data in the vegetation model and the asymmetric downscaling of damage to GWP in scenarios with high/low social equity capture part of the existing international differences. We do concede that we fail to take into account the heterogeneous impact on different sectors, but this was not the focus here. Overall, damage to GWP has a very minor effect on yield and cropland (Figure 6b) and therefore we do not see this as a critical shortcoming of the approach in the current study.

* More documentation of how the climate impacts were applied to the agricultural sectors should be provided. In this sort of vegetation and agro-economic model link-up, many countries typically see unrealistic and positive yield impacts, particularly places-sion paper with a harsh climate and low yields in the historical years, where small increases in precipitation can lead to large modeled yield increases. In my work with similar data I've had the most trouble with the Middle East, North Africa, Russia, and Canada. But to some extent this depends on the mathematical formulations for applying aggregated crop model output to the baseline nation-level yield trajectories.

Response: How crop yield outputs from LPJ-GUESS are fed into the land use model, PLUM, is described in detail in Engström et al. (2016b), and we will add a short summary in the revised manuscript. Current modelled yield levels are scaled to the actual yields synthesised by Mueller et al. (2012) and aggregated to country level. This is also true for the potential yields, and the difference between them is taken to be the current yield gap. Future yield levels are climate driven anomalies applied to the baseline levels, then aggregated based on the SSP specific land cover, simulated by PLUM, to country level. It is true that changes in precipitation can lead to increases in simulated yields but in the model this would not influence the relative yield gap, as both are affected arithmetically by the same climate driven yield increase.

* Next, I'll address a few of the simplifications and representations that struck me as

particularly problematic in the modeling exercise; unfortunately, without the raw data inputs and outputs to/from the model, I can really only guess as to the relative importance of each.

1) shareBEcr: this parameter, exogenous in all periods and scenarios, represents the combustible energy content of all ethanol and biodiesel feedstocks divided by total global bioenergy demands. The denominator includes all remaining uses of bioenergy, which the authors note account for some 97% of the base-year bioenergy demands. The basic problem is that these bioenergy commodities (in the denominator) have fundamentally different future demand drivers from ethanol and biodiesel (the numerator), so there isn't really any way to know a priori how this will evolve over time, in the various narratives of the SSPs. In the current study design, the authors are attempting to set the "shareBEcr" such that the quantity of agricultural crops used as bioenergy feedstocks does not grow by more than 30-50% from its base year value, according to the estimates of a study (Haberl et al. 2010). However, in the model, this is applied as a share constraint rather than a quantity constraint, so the target quantity (from Haberl et al. 2010) appears to be greatly exceeded in some if not all of the scenarios. On the other side, the bioenergy commodities that do grow a lot (up to 450 EJ/yr) are the unspecified ones, which in the study methods are not tied to cropland or the land/carbon models, even though it is stated that this commodity class includes ligno-cellulosic (i.e., "second-generation") bioenergy sources. These bioenergy crops are a very important component of future land use change in the SSP scenarios, and probably account for the vast majority ofndly version the growth of bioenergy here. This is because traditional uses of firewood and charcoal, and industrial recycling of bio-derived byproduct fuels, are simply not energy demands that are likely to scale up in any significant way in response to an emissions mitigation policy. So, by bunding second-generation bioenergy crops with waste and traditional biomass commodities whose production is not tied to land use, the scenarios are getting up to 450 EJ/yr of bioenergy, almost as high as total global primary energy consumption of all fuels today, without causing land use change or any other consquences relevant for emissions and carbon stocks.

Response: We admit that not explicitly modelling the production of second generation bioenergy feedstock is a shortcoming of our approach, and have added mention of this aspect to the limitation section in the discussion (4.3). In future work it would be desirable to include a wider range of bioenergy feedstocks. However, generally second generation feed stocks are typically either by-products or crops that contribute to carbon sequestration (e.g. switch grass, woody biomass).Therefor it is likely that the effect of increased bioenergy production on the carbon balance in our study is rather overestimated than underestimated, which is raised in the discussion in the revised manuscript. Generally, the estimates for the global technical potential of total bioenergy differ very largely in the literature, from e.g. < 300 EJ by 2050 (Erb et al., 2012; Haberl et al., 2011) to > 500 EJ by 2050 (Hoogwijk et al., 2005; Smeets et al., 2007). The lower range of the estimates typically includes food-first approaches and or sustainability constraints. However, given the nature of the different scenarios it is plausible that sustainable bioenergy potential are exceeded and bioenergy production occurs at higher environmental costs. In the revised manuscript we discuss unintended outcomes, such as a higher total bioenergy production in SSP1 mitigation scenario in light of the importance of a global carbon tax to avoid such outcomes. Also, due to the potential importance of lignocellulosic crops we assumed a lower contribution of energy crops to total bioenergy of max 15% in 2100 compared to Haberl et al., 2010 (30-50%). The assumed shares of crop-based bioenergy on total bioenergy for the SSPs are 3-9% (Table 4). A quantity share of 3-9% results in up to 11 EJ by 2050 and 47 EJ by 2100 of crop-based bioenergy, which is still below the (sustainable) potential estimated (e.g. 50% of 270 EJ by 2050 according to Harbel et al. 2010).

2) conversionEff: this parameter describes the relationship between the combustible energy content of harvested bioenergy and the biofuels produced, in the form of ethanol and biodiesels. The authors estimate this efficiency at about 65% in the base year, with a maximum value (year 2100, with efficiencyBEcrEJ set to 50%) of 95%. The end-of-century levels are simply not realistic; that would entail conversion processes wherein the vast majority of the combustible energy content of the by-products (dried

distillers grains and oil crop feedcakes) are somehow transferred into the fuel. I don't know what the theoretical limits on that conversion are, but I suspect it's closer to 65% than 95%.

Response: The documentation of conversionEff is clarified in the revised manuscript. As intended conversionEff improves as given in Table 4 (max 70% by 2100), which we would like to argue is a conservative approach.

3) A2 and A3: the annual improvement rate in the efficiency of producing coal and carbon-free energy, respectively. It is possible that this description is inaccurate in several ways; I'm hoping that what is intended is the improvement in the whole-economy energy intensity of the use of these fuels, or the ratio of primary (usable) energy to economic output. Improving the energy efficiency of producing these energy commodities (e.g., less fuel-intensive coal mining or farming practices) wouldn't make much difference to energy consumption at the global level, and in any case these practices are likely to become more energy-intensive over time, not less, due to resource depletion, mechanization of farming, and others. There are also problems if this were interpreted as the efficiency of using energy. An efficiency that grows at 2% per year from 2010 to 2100 ends up 6 times higher than it started, and for the maximum improvement rate used, 2.5%/yr, it ends up nearly 10 times higher. There are no uses of coal in the energy system, at a global level, with thermal efficiency levels low enough to permit this sort of improvement. And, like many parameters here, I would suggest calculating them from the model outputs in the publicly available SSP scenarios, and using some simplification from that calculation, rather than arbitrarily guessing. The SSP suggested parameterizations (guidelines) were written for IA models with a much higher level of detail of the physical systems than the tools used here.

Response: In the revised manuscript we clarified the definition of efficiency of producing coal and green energy describes the output of energy services per unit labour used in the respective energy sector (this is distinct from the overall energy efficiency in the economy (GPD per unit of energy) which is endogenously determined and from

thermal efficiency which obviously is bounded from above). In the US, coal production efficiency (coal produced per hour worked in coal production) increased by 3.2% per year between 1949 and 2011, this is about 1.2% more than general increases in labour productivity. For most decades the growth rate was substantially higher – if we disregard the 70s and the 00s, the average was 6.2% per year. (Source: US Energy Information Administration.) Thus, our assumptions are, broadly speaking, not out of line compared with historical growth rates. However, it is generally difficult and subject to much academic controversy to forecast future productivity growth (see e.g, the discussion between Robert Gordon and Andrew McAfee (Brynjolfsson and McAfee, 2013; Gordon, 2016)). Regarding green energy, it is arguably even more difficult. Thus, rather than settling for one estimate of future growth rates based on historic productivity, we provide a set of scenarios.

4) Yield: the yield growth rates I would also suggest taking from the SSP database, using area-weighted and indexed cereal yields in each region. The current method assigns baseline productivity growth on the basis of the yield gap, from the Mueller et al gridded yield gap study. There are two issues with this approach. For one, as the authors note, the rate at which countries close the yield gaps is tied to "each scenario's technological growth, economic development and technology transfer." However, these attributes are more granular than the inputs to the model used, and it isn't specified how those yield trajectories were developed. Second, convergence with base-year yield gaps is only one component of future agricultural productivity improvements; the distribution itself should also shift upwards due to technological change. In regions with no or little yield gap (e.g., Europe, the USA), yield improvements to 2100 are effectively frozen in this method, which likely isn't what is intended.

Response: As previously emphasised our intention was to produce independent SSP realisations and therefore we would like to refrain from calculating the yield growth rates from the SSP database. In the revised manuscript we specify the assumptions that lead to the yield trajectories (section 2.4.2). Admittedly, our approach is conservative insofar

as only the effect of climate change on yield growth is included, but not the use of new varieties (clarified in the revised manuscript). However, in countries with low yield gaps, current yields are close to the theoretical potential of vegetation growth, given current climate conditions.

5) p: the rate at which future welfare is discounted. Part of the problem with the research goals of this study is that the impacts of climate change from emissions today play out over hundreds of years, due to the long lifetime of CO2, not even taking into account issues like sea level rise or thermohaline cycle disruption. How the net present value of damages can be applied to an economy over such a long time span and across generations is a topic without consensus in the modern economic literature. Some review is warranted (e.g., Stern versus Nordhaus). Still, one point with good agreement is that the discount rate is very important for the balance between near-term emissions mitigation and long-term reduction in climate damages. I couldn't find where the discount rate was stated, but did find a statement that the discount rate was not varied in any sensitivity analysis, so I'd suggest clarifying what is used, stating the justification, and running a couple of sensitivity scenarios.

Response: The discount rate applied to future damages to GWP can be separated in one part that depends on how welfare in the future is valued relative to welfare today. The other part depends on the level of consumption in the future relative to today. A high relative level of future consumption reduces the relative value of a lost unit of future consumption due to lower marginal utility. The first part is often called the subjective discount rate and is purely determined by preferences/value judgements. Golosov et al (2014) show that under reasonable assumptions, the socially optimal carbon tax is independent of the other part of the discount rate, but highly sensitive to the first. In our approach we used a subjective discount rate of 1.5% per year (Nordhaus discount rate) which is stated in the revised manuscript. This is in accordance with estimates about how individuals discount over relatively short time-spans (up to a few decades, see e.g., (Nordhaus, 2007) ). Climate change operates over substantially longer horizons

and e.g., the Stern Review argues on moral grounds in favour of using substantially lower discount rates (Stern, 2007). In our model, the key consequence of using a lower discount rate is to increase the optimal tax and thus reduce emissions in the scenarios where the tax is used. For an indication of how sensitive the optimal tax is to changes in the discount rate, see Golosov et al. (2014)

—Specific items— p2 line 10 - mitigation isn't solely for the purpose of decreasing negative impacts on human society. also for terrestrial biosphere (e.g., biodiversity, ecosystem function).

Response: We added "and the terrestrial biosphere" in the revised manuscript.

p4 lines 6-8: climate impacts isn't the only factor driving yield changes over time (also yield gap convergence)

Response: We clarified the different drivers of yield changes in the revised manuscript.

p4 line _20: how are energy supplies modeled, in order to get supplies and demands to balance? Are there exogenous supply curves used?

Response: No, the supply and demand are determined by profit maximizing forms facing technological constraints and taxes.

p4 lines 20-21: all IA models represent energy markets explicitly, and have since the first-generation IA models back in the 1980's (e.g., Edmonds-Reilly-Barnes was first documented in 1986).

Response: The models referred to describe markets as supply and a demand functions calibrated to match historic data and this is very different from our approach (see response to first comment). The authors mentioned by the referee are quite explicit about this and note that calibrated supply and demand functions are not meant to be a reasonable description of human behaviour. The approach here is to make a description of the market participants, their objective functions, constraints and information. To reduce the risk of misunderstanding, we use the term "micro-foundations", in the re-
vised draft. We state that this is to be interpreted as "Supply and demand are derived from an explicit description of the objectives and constraints of forward-looking market participants operating in a potentially stochastic environment. ".

p4 lines 23-25: given the complexities involved, I don't see how one can reasonably state that the % GDP loss is a linear function of the global average temperature, but given that it is another study that is being cited, please provide a 1- 2 sentence description of how this was estimated in that study–over what time scale, geographic scale, temperature change, and was is an empirical estimate from historical data, or a model-derived estimate? It is crucially important for the results in this study, but strikes me as very questionable.

Response: This was not well described in the manuscript. We added description of this in the revised manuscript. It should be noted that the linearity is not from temperature to damages (this is convex) but from $CO_2$ concentration to the log of GWP.

p5 line 2: the emissions pathways from this model should be compared with the published ssp's, and harmonized to the extent possible.

Response: We will prepare a figure/table where emissions pathways of the different SSPs are compared with the results of Riahi et al (2017).

p5, lines 25-30: from my understanding of the methods later on, trade is set a priori and cropland expansion is used to modify the supply, so that it is equal to demand plus or minus net trade. this is a bit unusual in this field; in most models, trade is price sensitive, and can be an important determinant of the equlibrium between agricultural production and demand. it woud be a good idea to make sure the results from this approach are reasonable in India, which has already very high cropland shares, and a population that is growing fast and becoming more wealthy, both of which put significant upward pressure on agricultural product demands.

Response: The trade mechanism without explicitly modelling prices is one of the key

characteristics of the parsimonious land use model and has been evaluated for the time period 1991-2010 for selected countries. For India, the modelled cereal land captured the observed cereal land very well (Engström et al., 2016b).

p6, line 29: it is stated that bioenergy is only produced on abandoned cropland; what is used to estimate abandoned cropland? I'm not aware of any inventories that disaggregate this quantity specifically, but there are vast quantities of land in the former Soviet Union (Central Asia), the Middle East, and the forests of the eastern United States that were cropland at some point in human history. it is hard to see how these lands would be the preferred sites for bioenergy production, particularly in light of the locations where cropland expansion is currently taking place (e.g., tropical rainforests).

Response: What was meant here was that in first place bioenergy production is simulated to occur in countries where yield improvements (and/or decreasing demand) freed cropland in the previous time-step. We clarified this in the revised manuscript.

p7 line 30: the Hurtt et al (2011) dataset distinguished pasture on the basis of land use, not land cover class. it classified as pasture vast tracts of land area that are not grassland, including most of Tibet, Australia, Central Asia, and the western USA. it's probably not correct to assume this is all grass, but it might also not be important for the study; I can't tell.

Response: It is true that the pasture areas from Hurtt et al. (2011) cover large areas that should probably not be considered as pastures, but rather as rangelands. This is not explicitly covered in the model, but the implemented management (grazing and cutting) of the pastures in LPJ-GUESS is an intermediate between intensive and extensive. Also, climatic conditions and soils will influence the productivity and carbon sequestration locally, and thus capture e.g. the low productivity in the areas where the land cover is rangelands rather than pastures.

p8 line 10: irrigation, N application, and tillage intensity are held at base year levels while yield gaps are assumed to close. However, in Mueller et al. (2012), these were

the main factors that account for present-day yield gaps.

Response: It is a limitation of our study that we account only for impacts of biophysical forcings, i.e. climate change, CO2 fertilisation and N deposition on yield gap, not effects of management changes such as irrigation, N application and tillage. This will be pointed out in a revised version of the Limitations subsection (4.3) in the discussion. It should be noted that the cropland yield simulations are not used directly in PLUM, but are used to calculate a country-specific scaling factor for the yield gap. This will be made more clear in the revised manuscript where we will add a short description of the yield gap implementation in PLUM (see also reply to comment above).

p11 - for any grid cell, the yield impact is not a simple linear function of the radiative forcing. I'm not sure what is gained by using this probability-weighted approach as opposed to just simply assigning a single RCP scenario that is most similar to the emissions outcome of the given scenario.

Response: Using the probability-weighted approach has the advantage that in case of a concentration scenario of a given SSP that lie between two RCPs we did not need to favour one RCP over the other or simulate a larger number of scenarios (multiple combinations). This is clarified in the revised manuscript (p x, l. y).

Figure 4 - Please clarify whether global cropland (4d) includes global cropland for bioenergy (4c). It did in the SSP reporting database and in Schmitz et al. (2014), so hopefully it does here too!

Response: Yes, global cropland includes global cropland for bioenergy, which is clarified in the revised manuscript.

References Ahlström, A., Schurgers, G., Arneth, A., and Smith, B.: Robustness and uncertainty in terrestrial ecosystem carbon response to CMIP5 climate change projections, Environmental Research Letters, 7, 044008, 2012. Bruckner, T., Bashmakov, I. A., Mulugetta, Y., Chum, H., de la Vega Navarro, A., Edmonds, J., Faaij, A., Fungtammasan, B., Garg, A., Hertwich, E., Honnery, D., Infield, D., Kainuma, M., Khennas, S., Kim, S., Nimir, H. B., Riahi, K., Strachan, N., Wiser, R., and Zhang, X.: Energy Systems. In: Climate Change 2014: Mitigation of Climate Change. Contribution of Working Group III to the Fifth Assessment Report of the Intergovernmental Panel on Climate Change [Edenhofer, O., R. Pichs-Madruga, Y. Sokona, E. Farahani, S. Kadner, K. Seyboth, A. Adler, I. Baum, S. Brunner, P. Eickemeier, B. Kriemann, J. Savolainen, S. Schlömer, C. von Stechow, T. Zwickel and J.C. Minx (eds.)]. Cambridge University Press, Cambridge, United Kingdom and New York, NY, USA., 2014. Brynjolfsson, E. and McAfee, A.: The Second Machine Age (2013), , WW Norton Co, 2013. Engström, K., Olin, S., Rounsevell, M. D. A., Brogaard, S., van Vuuren, D. P., Alexander, P., Murray-Rust, D., and Arneth, A.: Assessing uncertainties in global cropland futures using a conditional probabilistic modelling framework, Earth System Dynamics Discussions, 1-33, 2016a. Engström, K., Rounsevell, M. D. A., Murray-Rust, D., Hardacre, C., Alexander, P., Cui, X., Palmer, P. I., and Arneth, A.: Applying Occam's razor to global agricultural land use change, Environmental Modelling & Software, 75, 212-229, 2016b. Erb, K.-H., Haberl, H., and Plutzar, C.: Dependency of global primary bioenergy crop potentials in 2050 on food systems, yields, biodiversity conservation and political stability, Energy Policy, 47, 260-269, 2012. Friedlingstein, P., Meinshausen, M., Arora, V. K., Jones, C. D., Anav, A., Liddicoat, S. K., and Knutti, R.: Uncertainties in CMIP5 Climate Projections due to Carbon Cycle Feedbacks, Journal of Climate, 27, 511-526, 2014. Golosov, M., Hassler, J., Krusell, P., and Tsyvinski, A.: Optimal Taxes on Fossil Fuel in General Equilibrium, Econometrica, 82, 41-88, 2014. Gordon, R. J.: The Rise and Fall of American Growth: The U.S. Standard of Living Since the Civil War, 2016. Haberl, H., Beringer, T., Bhattacharya, S. C., Erb, K.-H., and Hoogwijk, M.: The global technical potential of bio-energy in 2050 considering sustainability constraints, Current Opinion in Environmental Sustainability, 2, 394-403, 2010. Haberl, H., Erb, K.-H., Krausmann, F., Bondeau, A., Lauk, C., Müller, C., Plutzar, C., and Steinberger, J. K.: Global bioenergy potentials from agricultural land in 2050: Sensitivity to climate change, diets and yields, Biomass and Bioenergy, 35, 4753-4769, 2011. Hoogwijk,

[Figure]

M., Faaij, A., Eickhout, B., Devries, B., and Turkenburg, W.: Potential of biomass energy out to 2100, for four IPCC SRES land-use scenarios, Biomass and Bioenergy, 29, 225-257, 2005. Hurtt, G. C., Chini, L. P., Frolking, S., Betts, R. A., Feddema, J., Fischer, G., Fisk, J. P., Hibbard, K., Houghton, R. A., Janetos, A., Jones, C. D., Kindermann, G., Kinoshita, T., Klein Goldewijk, K., Riahi, K., Shevliakova, E., Smith, S., Stehfest, E., Thomson, A., Thornton, P., Vuuren, D. P., and Wang, Y. P.: Harmonization of land-use scenarios for the period 1500–2100: 600 years of global gridded annual land-use transitions, wood harvest, and resulting secondary lands, Climatic Change, 109, 117-161, 2011. Lindeskog, M., Arneth, A., Bondeau, A., Waha, K., Seaquist, J., Olin, S., and Smith, B.: Implications of accounting for land use in simulations of ecosystem carbon cycling in Africa, Earth System Dynamics, 4, 385-407, 2013. Mueller, N. D., Gerber, J. S., Johnston, M., Ray, D. K., Ramankutty, N., and Foley, J. A.: Closing yield gaps through nutrient and water management, Nature, 490, 254-257, 2012. Murray-Tortarolo, G., Anav, A., Friedlingstein, P., Sitch, S., Piao, S., Zhu, Z., Poulter, B., Zaehle, S., Ahlström, A., Lomas, M., Levis, S., Viovy, N., and Zeng, N.: Evaluation of land surface models in reproducing satellite-derived LAI over the high-latitude Northern Hemisphere: Uncoupled DGVMs, Remote Sensing, 5, 4819-4838, 2013. Nordhaus, W.: A Question of Balance: Weighing the Options on Global Warming Policies, Yale University Press, New Haven, CT, 2008. Nordhaus, W.: A Review of the Stern Review on the Economics of Climate Change, A Review of the Stern Review on the Economics of Climate Change, XLV, 686-702, 2007. Piao, S., Sitch, S., Ciais, P., Friedlingstein, P., Peylin, P., Wang, X., Ahlström, A., Anav, A., Canadell, J. G., Cong, N., Huntingford, C., Jung, M., Levis, S., Levy, P. E., Li, J., Lin, X., Lomas, M. R., Lu, M., Luo, Y., Ma, Y., Myneni, R. B., Poulter, B., Sun, Z., Wang, T., Viovy, N., Zaehle, S., and Zeng, N.: Evaluation of terrestrial carbon cycle models for their response to climate variability and to $CO_2$ trends., Global Change Biology 19, 2117-2132, 2013. Riahi, K., van Vuuren, D. P., Kriegler, E., Edmonds, J., O'Neill, B. C., Fujimori, S., Bauer, N., Calvin, K., Dellink, R., Fricko, O., Lutz, W., Popp, A., Cuaresma, J. C., Kc, S., Leimbach, M., Jiang, L., Kram, T., Rao, S., Emmerling, J., Ebi, K., Hasegawa, T., Havlik, P., Humpenöder, F., Da
Silva, L. A., Smith, S., Stehfest, E., Bosetti, V., Eom, J., Gernaat, D., Masui, T., Rogelj, J., Strefler, J., Drouet, L., Krey, V., Luderer, G., Harmsen, M., Takahashi, K., Baumstark, L., Doelman, J. C., Kainuma, M., Klimont, Z., Marangoni, G., Lotze-Campen, H., Obersteiner, M., Tabeau, A., and Tavoni, M.: The Shared Socioeconomic Pathways and their energy, land use, and greenhouse gas emissions implications: An overview, Global Environmental Change, 42, 153-168, 2017. Shevliakova, E., Pacala, S. W., Malyshev, S., Hurtt, G. C., Milly, P. C. D., Caspersen, J. P., Sentman, L. T., Fisk, J. P., Wirth, C., and Crevoisier, C.: Carbon cycling under 300 years of land use change: Importance of the secondary vegetation sink, Global Biogeochemical Cycles, 23, n/a-n/a, 2009. Smeets, E., Faaij, A., Lewandowski, I., and Turkenburg, W.: A bottom-up assessment and review of global bio-energy potentials to 2050, Progress in Energy and Combustion Science, 33, 56-106, 2007. Smith, B., Perentice, I.C., Sykes, M.T.: Representation of vegetation dynamics in the modelling of terrestrial ecosystems: comparing two contrasting approaches within European climate space, Global Ecology and Biogeography, 10, 621-637, 2001. Smith, B., Wårlind, D., Arneth, A., Hickler, T., Leadley, P., Siltberg, J., and Zaehle, S.: Implications of incorporating N cycling and N limitations on primary production in an individual-based dynamic vegetation model, Biogeosciences, 11, 2027-2054, 2014. Stern, N. H.: The Economics of Climate Change: The Stern Review, Cambridge and New York: Cambridge University Press, 2007.

---

## Author Comment (AC2) · 3 Mar 2017

Resonse to D.B. Kirk-Davidoff (Referee)

As editor, I am submitting a reviewer comment, thus closing the discussion, in light of the authors' long wait for the completion of this review process and of the first reviewer's excellent and thorough review. The authors present a study of an integrated assessment model in which they first find parameter settings that allow the model to approximate a set of scenarios described in the Shared Socio-economic Pathways framework, and then add a mechanism intended to represent a carbon tax imposed on fossil fuel combustion, and note the impact of this tax on gross world product, on fossil fuel use, and on agricultural activity. I concur with the first reviewer's comments

and recommendations, especially with regard to the desirability of a more realistic accounting of the local cost to wealth of climate impacts (since these are already spatially resolved for use in the ecosystem model). I have a number of additional suggestions for clarifications. It was not immediately clear to me that the various SSP scenarios would be imposed on the model through parameter choices (as opposed, for example to forcing the model through variable population growth rates or some other forcing mechanism). The introduction could be rewritten to make this much clearer, and also to address the first reviewer's concerns about how consistent the model trajectories are with the SSPs as defined. The discussion of "damage on GWP" is confusing- this seems to be just a proxy for global warming averted, but since the climate-economy model calculates GWP explicitly, couldn't the GWP itself be shown, so that the increase in GWP due to the optimized carbon tax would be apparent in Figure 4? Similarly, the terms "challenges to adaptation" and "challenges to mitigation" don't seem to be as parallel in meaning as their grammatical parallelism would suggest. "Challenges to adaptation" seems to indicate political resistance to adaptation, while "challenges to adaptation" seems to indicate a structural likelihood of a lot of damage to GWP due to warming. Perhaps these should be rephrased as "resistance to mitigation" and "wealth available for adaptation" (which would have the opposite sign to "challenges to adaptation")? In this regard, l. 21 on page 8 seems problematic without a clear baseline: since higher means more damage per unit Carbon emitted, the required reduction in emissions to achieve a given reduction in damage is actually less, though of course the reduction in emissions required to achieve a given low *level* of damage would be larger.

Response: Dear editor, Thank you for your comments. The model is driven by inputs such as population and economic growth and scenarios are differentiated on the basis of these inputs, as well as parameter choices. Following your suggestion this is clarified in the introduction, also emphasising the value of the study in providing independent interpretations of the SSP narratives, compared with the published realisations in the SSP database. It is important to note there is no objective "right" interpretation of these

(qualitative and relative) narratives in quantitative terms. Thus there is a place for different interpretations based on different defendable approaches to stimulate debate and accommodate the many dimensions of uncertainty surrounding the actual evolution of the biophysical-societal system in the 21st century. In this light we believe our study is a relevant and valid alternative contribution, complementing the "official" SSP projections. The damage on GWP is proxy for the impact of global warming. The climate economy model does not calculate GWP explicitly, but GWP is the sum of all countries' GDPs (provided from the SSP database). Throughout the manuscript we used the SSP terminology which 'collapses' the multi-faceted futures described by each SSP onto the two dimensions of "challenges to adaptation" and "challenges to mitigation". This terminology is now well-established in relevant literature and it would be confusing to adopt alternative terminology in this paper. We assumed that the challenge to mitigation impacts the level of the global carbon tax that can be implemented in the respective scenario. For example, with larger challenges to mitigation the carbon tax level would be less optimal, while the level of the global carbon tax is optimal in scenarios with low challenges to mitigation. We clarified in the revised manuscript that the characteristics of the SSPs determine the challenge to mitigation and that it is not a political resistance to mitigation per se that differentiates the SSPs. For example, in the regionalized, not internationally cooperating SSP3 world with the use of unconventional and domestic energy resources the implementation of a global optimal carbon tax would be very challenging. Challenges to adaptation map to both the level of climate-related damages expected in the absence of adaptation, and the amount of adaptive capacity implied by the SSP storyline. Thus, for example, the highly engineered and developed infrastructure and attainment of human development goals implies a low challenge for adaptation, rather that there is much wealth needed for adaptation.